# PLK4 promotes centriole duplication by phosphorylating STIL to link the procentriole cartwheel to the microtubule wall

Tyler Chistopher Moyer, Andrew Jon Holland*

Department of Molecular Biology and Genetics, Johns Hopkins University School of Medicine, Baltimore, United States

**Abstract** Centrioles play critical roles in organizing the assembly of the mitotic spindle and templating the formation of primary cilia. Centriole duplication occurs once per cell cycle and is regulated by Polo-like kinase 4 (PLK4). Although significant progress has been made in understanding centriole composition, we have limited knowledge of how PLK4 activity controls specific steps in centriole formation. Here, we show that PLK4 phosphorylates its centriole substrate STIL on a conserved site, S428, to promote STIL binding to CPAP. This phospho-dependent binding interaction is conserved in *Drosophila* and facilitates the stable incorporation of both STIL and CPAP into the centriole. We propose that procentriole assembly requires PLK4 to phosphorylate STIL in two different regions: phosphorylation of residues in the STAN motif allow STIL to bind SAS6 and initiate cartwheel assembly, while phosphorylation of S428 promotes the binding of STIL to CPAP, linking the cartwheel to microtubules of the centriole wall.

DOI: https://doi.org/10.7554/eLife.46054.001

*For correspondence: aholland@jhmi.edu

Competing interests: The authors declare that no competing interests exist.

## Introduction

Centrioles are microtubule-based structures that recruit a surrounding pericentriolar material (PCM) to form the centrosome (*Nigg and Holland, 2018*; *Gönczy, 2012*). Centrosomes nucleate the formation of the microtubule cytoskeleton in interphase cells and form the poles of the mitotic spindle during cell division. In quiescent cells, centrioles dock at the plasma membrane and act as basal bodies that template the formation of cilia and flagella (*Breslow and Holland, 2019*). Cycling cells tightly couple centriole biogenesis with cell cycle progression. Centriole duplication begins at the G1-S phase transition when a new procentriole grows perpendicularly from a single site at the proximal end of each of the two parent centrioles. In late G2 phase, the two centriole pairs separate and increase PCM recruitment to promote the formation of the mitotic spindle. At the end of mitosis, the centrosomes are equally partitioned so that each daughter cell inherits a pair of centrioles. Defects in centriole biogenesis can result in the formation of supernumerary centrosomes which promote mitotic errors that can contribute to tumorigenesis (*Levine et al., 2017*; *Levine and Holland, 2018*; *Basto et al., 2008*; *Serçin et al., 2016*; *Coelho et al., 2015*). Moreover, mutations in centriole proteins are linked to growth retardation syndromes and autosomal recessive primary microcephaly (MPCH) in human patients (*Nigg and Raff, 2009*; *Chavali et al., 2014*).

The initiation of centriole duplication requires a conserved set of five core proteins: PLK4 (ZYG-1 in *C. elegans*), CEP192 (SPD-2 in *C. elegans* and Spd-2 in *Drosophila*), CPAP (also known as CENPJ, SAS-4 in *C.elegans* and Sas-4 in *Drosophila*), STIL (SAS-5 in *C. elegans* and Ana-2 in *Drosophila*) and SAS6 (*Leidel et al., 2005*; *Leidel and Gönczy, 2003*; *Dammermann et al., 2004*; *Delattre et al., 2004*; *Kemp et al., 2004*; *O'Connell et al., 2001*; *Pelletier et al., 2004*; *Kirkham et al., 2003*). Of

**eLife digest** A cell's DNA is the chemical instruction manual for everything it does. Each cell in our bodies contains over two meters of DNA, which is divided into 46 packages of information called chromosomes. When the body needs to make more cells, for example during growth or repair, existing cells divide in two in order to replicate themselves. This means that they also need to copy all of their DNA and then deliver identical sets of chromosomes to each new cell.

Animal cells use structures called centrioles to help them divide their sets of chromosomes accurately. When cells are about to divide, they make a new set of centrioles by assembling a variety of proteins. This assembly process must be carefully controlled; if too many or too few centrioles are built, cell division errors can occur that lead to the generation of new cells with abnormal numbers of chromosomes.

The enzyme PLK4 helps to assemble centrioles, but its exact role in the construction process has remained largely unknown. For example, how it might modify different components of the centriole, and why this matters, is poorly understood.

By performing cell biological and biochemical experiments using human cells, Moyer and Holland show that PLK4 interacts with a protein called STIL that is found in the central part of the centriole. The modification of STIL at a specific location by PLK4 was needed to link it to another protein in the outer wall of the centriole, and was also necessary for the cells to build new centrioles. Cells in which PLK4 was unable to modify STIL had too few centrioles when they were beginning to divide.

Testing the activity of PLK4 in fruit flies revealed that it plays a similar role as in human cells. This suggests that the modification of STIL by PLK4 is important for normal cell division across different species.

The results presented by Moyer and Holland help us to understand how dividing cells build the complex machinery that enables them to pass on their genetic material accurately. Future work that builds on these findings could provide insight into human diseases, such as brain development disorders and cancer, where centrioles are either defective or present in the wrong number.

DOI: https://doi.org/10.7554/eLife.46054.002

these components, PLK4 has been identified as the central regulator of centriole assembly (*Habedanck et al., 2005*; *Bettencourt-Dias et al., 2005*). In mammalian cells, PLK4 is recruited to the centriole during G1 phase through binding to its centriole receptors CEP152 and CEP192, which encircle the proximal end of the parent centriole (*Cizmecioglu et al., 2010*; *Hatch et al., 2010*; *Kim et al., 2013*; *Park et al., 2014*; *Sonnen et al., 2013*). At G1-S phase transition, PLK4 transforms from a ring-like localization to a single focus on the wall of the parent centriole that marks the site of procentriole formation (*Kim et al., 2013*; *Ohta et al., 2014*; *Sonnen et al., 2012*; *Dzhindzhev et al., 2017*). This transition bears features of a symmetry breaking reaction and can be recreated in silico using two positive feedback loops that act on PLK4 (*Goryachev and Leda, 2017*; *Leda et al., 2018*). Binding of PLK4 to its centriole substrate STIL promotes activation of the kinase and is required for its ring-to-dot transformation (*Ohta et al., 2014*; *Moyer et al., 2015*; *Lopes et al., 2015*). PLK4 phosphorylates STIL in a conserved STAN motif to promote the binding and recruitment of SAS6 (*Ohta et al., 2014*; *Moyer et al., 2015*; *Dzhindzhev et al., 2014*; *Kratz et al., 2015*). SAS6 homo-oligomerizes to organize the central cartwheel, a stack of ring-like assemblies with nine-fold symmetry that provides the structural foundation for the procentriole (*Kitagawa et al., 2011a*; *van Breugel et al., 2011*; *van Breugel et al., 2014*; *Cottee et al., 2015*; *Guichard et al., 2017*).

Following cartwheel assembly, the centriole protein CPAP plays a critical role in the formation and stabilization of the triplet microtubule blades that make up the procentriole wall. CPAP interacts with multiple centriole proteins including STIL (*Cottee et al., 2013*; *Tang et al., 2011*; *Vulprecht et al., 2012*), CEP152 (*Cizmecioglu et al., 2010*; *Dzhindzhev et al., 2010*), CEP120 (*Lin et al., 2013a*), CEP135 (*Lin et al., 2013b*), and Centrobin (*Gudi et al., 2015*). The C-terminal TCP domain of CPAP directly binds to a highly conserved PRP motif in STIL (*Cottee et al., 2013*; *Tang et al., 2011*; *Vulprecht et al., 2012*), while the N-terminal domain of CPAP interacts with α/β-tubulin heterodimers (*Sharma et al., 2016*; *Zheng et al., 2016*; *Hung et al., 2004*). These

interactions allow CPAP to act as a molecular link between the cartwheel and the triplet microtubule blades of the centriole wall. Importantly, an MCPH mutation in the CPAP TCP domain weakens the STIL-CPAP interaction, highlighting the importance of this complex in centriole assembly and function (*Cottee et al., 2013*; *Tang et al., 2011*; *Bond et al., 2005*). CPAP positively regulates centriolar microtubule growth and, consequently, overexpression of CPAP leads to the formation of overly long centrioles in human cells (*Tang et al., 2009*; *Schmidt et al., 2009*; *Kohlmaier et al., 2009*). In addition to its role in controlling microtubule growth, CPAP also functions in recruiting PCM, either through the direct tethering of PCM proteins or by recruiting Plk1/Polo which is critical in promoting PCM assembly (*Zheng et al., 2014*; *Gopalakrishnan et al., 2011*; *Chou et al., 2016*; *Novak et al., 2016*).

At present, the binding of SAS6 to the STAN motif of STIL is the only interaction known to be controlled by PLK4 kinase activity. While the regulation of this assembly step is conserved in humans and flies (*Ohta et al., 2014*; *Moyer et al., 2015*; *Dzhindzhev et al., 2014*; *Kratz et al., 2015*), it is unclear whether this takes place in *C. elegans*, where ZYG-1/PLK4 directly binds and recruits SAS6 to promote cartwheel assembly (*Lettman et al., 2013*). Moreover, although phosphomimetic mutations in the crucial PLK4 phosphorylation sites in the STAN motif of STIL are functional, they cannot support centriole duplication in the absence of PLK4 kinase activity (*Kim et al., 2016*). This suggests that PLK4 must phosphorylate STIL, or other substrates, at additional sites to promote centriole assembly. Indeed, recent work in *Drosophila* identified additional PLK4 phosphorylation sites required for centriole biogenesis in the N-terminus of Ana2/STIL, but exactly how these phosphorylation events contribute to centriole formation remains unclear (*Dzhindzhev et al., 2017*; *McLamarrah et al., 2018*).

In this manuscript, we identify a conserved PLK4 phosphorylation site on STIL that promotes binding to CPAP in vitro and in vivo. This phospho-dependent binding interaction is conserved in flies and allows STIL to link the growing cartwheel to the outer microtubule wall of the centriole. Together, our findings offer insight into a novel step in centriole assembly that is regulated by PLK4 kinase activity.

## Results

### PLK4 phosphorylates STIL to promote CPAP binding

PLK4 phosphorylates conserved residues in the STIL STAN motif to promote binding to SAS6 (*Ohta et al., 2014*; *Moyer et al., 2015*; *Dzhindzhev et al., 2014*). To determine whether phosphorylation of STIL by PLK4 might affect the interaction of STIL with other components of the centriole duplication machinery, we tested the ability of Myc-GFP-STIL to interact with its known centriolar binding partners in the presence of kinase active (PLK4$^{WT}$) or kinase dead (PLK4$^{KD}$) PLK4. Active PLK4 triggers its own degradation and thus, we used a PLK4$^{\Delta24}$ mutant that stabilizes the active kinase by preventing PLK4-induced autodestruction (*Holland et al., 2010*). Expression of kinase active PLK4$^{\Delta24}$-mCherry increased the binding of STIL to SAS6 in cells (*Figure 1A*), but did not increase binding to the STIL-interacting partners RTTN (*Chen et al., 2017*) or CEP85 (*Figure 1B,C*) (*Liu et al., 2018*). Unexpectedly, we observed that PLK4 kinase activity promoted a robust increase in STIL binding to CPAP, suggesting that PLK4 kinase activity also controls the interaction of CPAP with STIL (*Figure 1D*).

To determine how PLK4 phosphorylation promotes binding of CPAP to STIL, we mapped in vitro PLK4 phosphorylation sites on STIL using mass spectrometry. Recombinant full-length GST-STIL was phosphorylated with the His-PLK4 kinase domain in vitro. Of the 84 in vitro phosphorylation sites we identified on STIL, S428 was of particular interest as it is highly conserved, matches the PLK4 consensus phosphorylation sequence and is positioned close to the known CPAP binding region on STIL (*Figure 2A*, *Figure 2—figure supplement 1*) (*Cottee et al., 2013*; *Kettenbach et al., 2012*; *Johnson et al., 2007*; *Hatzopoulos et al., 2013*). To determine if phosphorylation of STIL S428 was responsible for enhancing the binding of CPAP to STIL, we co-expressed FLAG-CPAP and a wild type (WT) or S428A mutant of Myc-GFP-STIL in the presence of kinase active or inactive PLK4$^{\Delta24}$-mCherry. The expression of kinase active PLK4 promoted a > 7 fold increase in the amount of CPAP bound to WT STIL, but this increased binding was not observed with STIL S428A (*Figure 2B*). To test if this phospho-regulated binding interaction can be reconstituted with purified components,

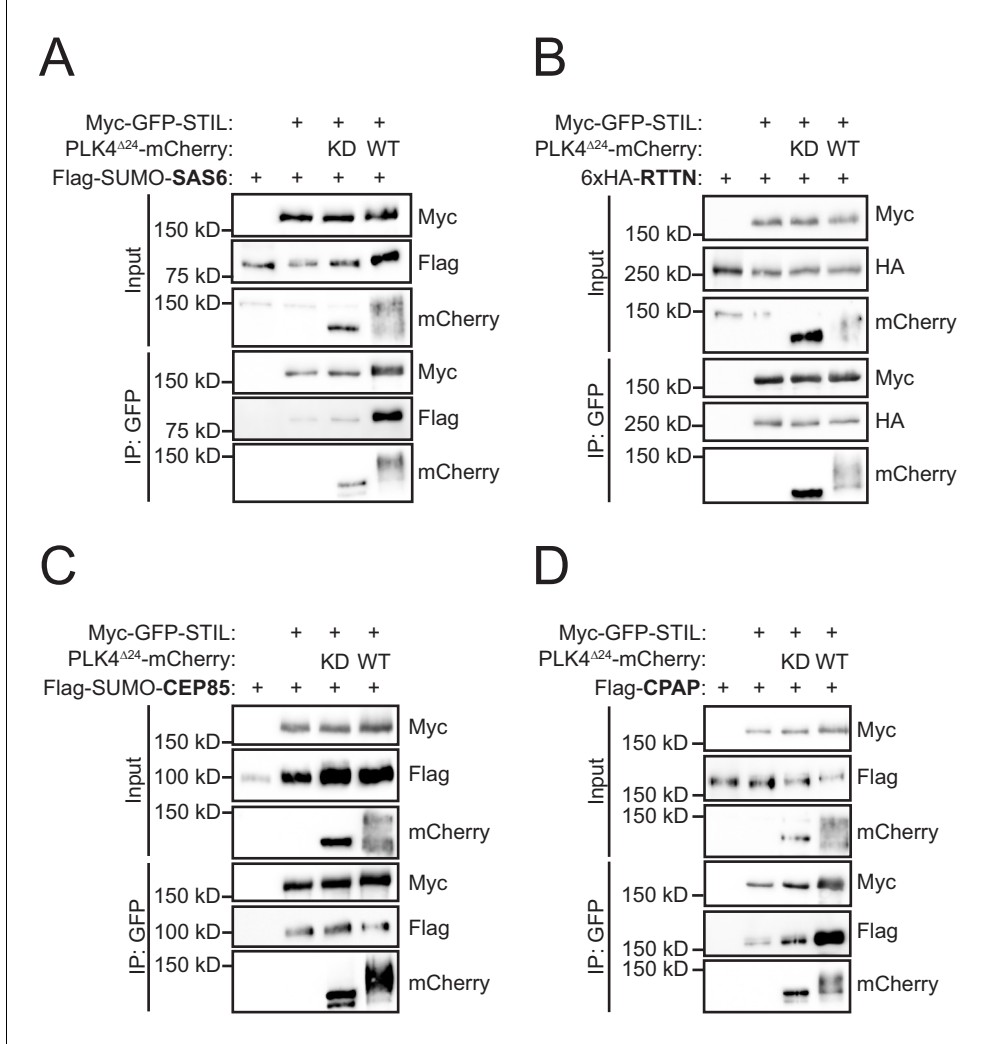

**Figure 1.** PLK4 kinase activity promotes STIL binding to CPAP. (**A–D**) HEK293FT cells were transfected with the indicated constructs, subjected to co-immunoprecipitation and immunoblotted with the indicated antibodies. PLK4 activity increased binding of both SAS6 and CPAP to STIL.

DOI: https://doi.org/10.7554/eLife.46054.003

we performed GST-pull down experiments on recombinant WT or S428A GST-STIL that had been phosphorylated with the His-PLK4 kinase domain and then incubated with a recombinant Flag-CPAP TCP domain. Phosphorylation of WT GST-STIL with PLK4 increased binding to the Flag-TCP domain by ~2.5 fold, but this increased binding was not observed with STIL S428A (*Figure 2C*). The use of the CPAP TCP domain rather than full-length protein may explain the more modest increase in CPAP binding to STIL in vitro compared to in vivo. These data show that phosphorylation of STIL S428 promotes CPAP binding to STIL in vitro and in vivo.

To demonstrate STIL S428 is a *bona fide* PLK4 phosphorylation site, we raised a phospho-specific antibody to this site. The affinity-purified pS428 antibody recognized recombinant GST-STIL in the presence of ATP and the His-PLK4 kinase domain, but not in the absence of ATP (*Figure 2D*). More-over, recognition of phosphorylated GST-STIL by the pS428 antibody was abolished by the S428A mutation, demonstrating the specificity of the pS428 antibody (*Figure 2D*). To determine if PLK4 can phosphorylate STIL S428 in cells, we co-expressed WT or a S428A mutant of Myc-GFP-STIL with kinase active or inactive PLK4$^{\Delta 24}$-mCherry (*Figure 2E*). The pS428 antibody recognized WT STIL in the presence of kinase active, but not kinase inactive, PLK4, showing that PLK4 phosphorylates STIL at S428 in vitro and in vivo.

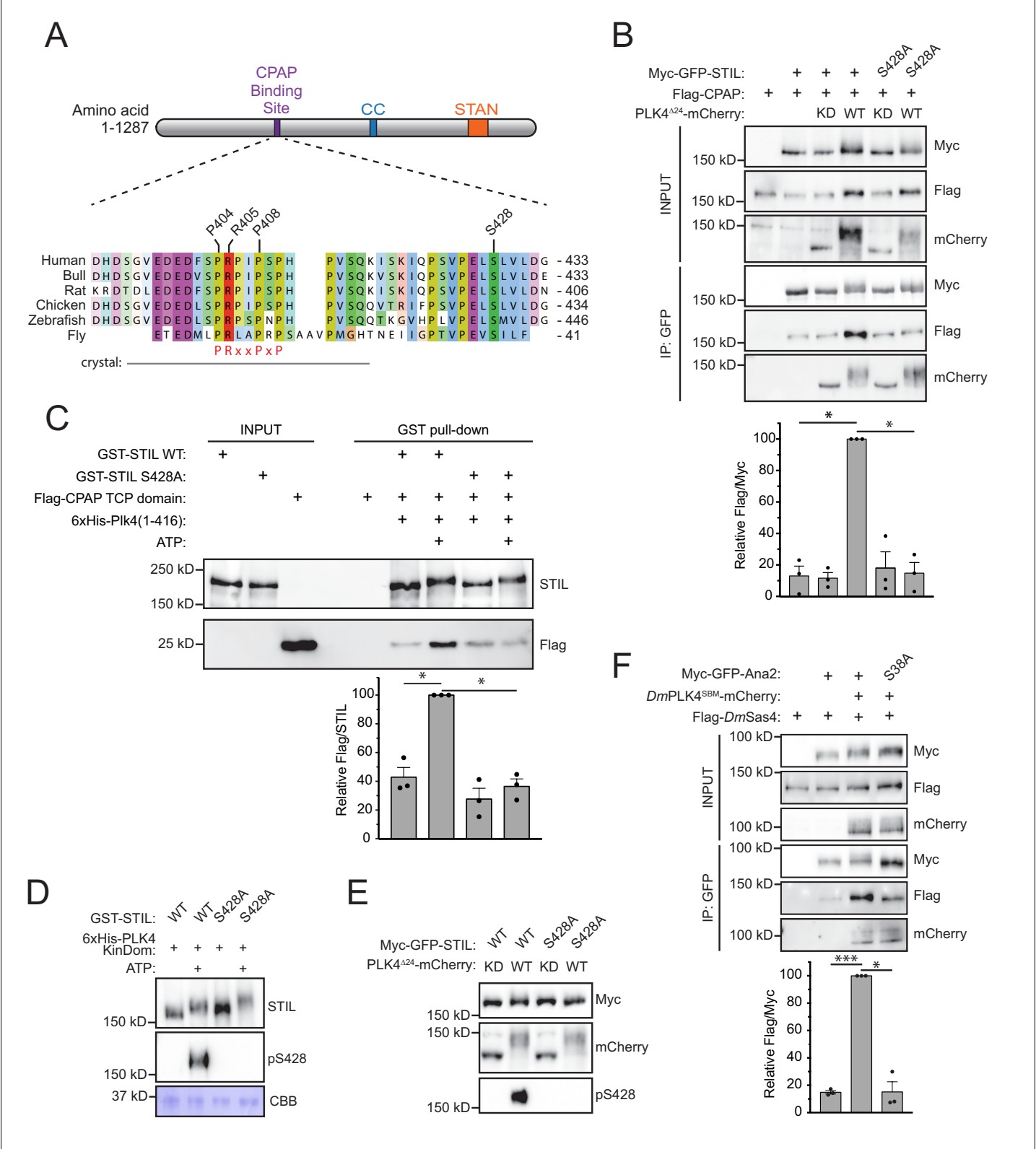

**Figure 2.** Phosphorylation of STIL by PLK4 promotes increased CPAP binding. (**A**) Schematic of full-length STIL showing locations of the known CPAP binding motif, coiled-coil (CC), and STAN motif important for SAS6 binding. Sequence alignment shows the conserved CPAP binding motif. The fragment of STIL crystalized with CPAP is shown by the gray bar (***Cottee et al., 2013***). STIL residues key to the STIL-CPAP interaction (P404, R405, and P408) are highlighted. The conserved PRxxPxP motif is indicated in red. Alignment was generated using Muscle under standard parameters. (**B**)
*Figure 2 continued on next page*

*Figure 2 continued*

HEK293FT cells were transfected with the indicated constructs, subjected to co-immunoprecipitation and immunoblotted with the indicated antibodies. The graph represents the mean of relative levels of immunoprecipitated Flag/Myc signal across three independent experiments. A dot displays measurements from each experiment. Error bars represent the standard error of the mean. (C) Recombinant full-length GST-STIL (WT or S428A) was bound to beads, phosphorylated by recombinant 6xHis-PLK4 kinase domain, and then incubated with Flag-CPAP TCP domain. GST-pulldowns were analyzed by immunoblotting with the indicated antibodies. The graph represents the mean of relative levels of Flag-TCP pulled down with GST-STIL. A dot displays measurements from each experiment. Error bars represent the standard error of the mean. (D) Recombinant full-length GST-STIL (WT or S428A) was phosphorylated in vitro by recombinant 6xHis-PLK4 kinase domain. Samples were immunoblotted with the indicated antibodies. PLK4 phosphorylates STIL at multiple sites leading to a reduced mobility on an SDS-PAGE gel. CBB represents 'coomassie brilliant blue' staining and shows the recombinant PLK4 kinase domain. (E) HEK293FT cells were transfected with the indicated constructs and immunoblotted with the indicated antibodies. (F) *Drosophila melanogaster* S2 cells were transfected with the indicated constructs, subjected to co-immunoprecipitation and immunoblotted with the indicated antibodies. *Dm*PLK4^SBM (Slimb Binding Mutant) is stabilized through mutation of critical serines in the Slimb-binding domain to alanine, which prevents auto-regulation of the kinase (*Rogers et al., 2009*). The graph represents the mean of relative levels of immunoprecipitated Flag/Myc signal across three independent experiments. A dot displays measurements from each experiment. Error bars represent the standard error of the mean. Asterisks indicate statistically significant differences between measurements (*: p<0.05; **: p<0.005; ***: p<0.0005). Statistics for all figures were calculated using a one-sample t-test where mean values were tested as being different from a value of 100.

DOI: https://doi.org/10.7554/eLife.46054.004

The following source data and figure supplement are available for figure 2:

**Source data 1.** *Figure 2* Data and Statistical Analysis.
DOI: https://doi.org/10.7554/eLife.46054.006
**Figure supplement 1.** In vitro PLK4 phosphorylation sites on STIL.
DOI: https://doi.org/10.7554/eLife.46054.005

## The phosphorylation-dependent binding of CPAP to STIL is conserved in flies

*D. melanogaster* PLK4 (DmPLK4) was recently shown to phosphorylate the STIL homolog Ana2 at S38, a residue equivalent to S428 in the human STIL (*Dzhindzhev et al., 2017*; *McLamarrah et al., 2018*) (*Figure 2A*). Phosphorylation of S38 was shown to be required for Ana2 recruitment (*Dzhindzhev et al., 2017*) but was reported to not alter Ana2 binding to the CPAP orthologue DmSas4 (*McLamarrah et al., 2018*). We reasoned that this discrepancy might arise because PLK4 is a low-abundance protein that is activated at the centriole and is unable to efficiently phosphorylate a significant fraction of the transfected Ana2. To test this possibility, we transfected *D. melanogaster* S2 cells with either WT or S38A Myc-GFP-Ana2 and Flag-DmSas4 in the presence or absence of a stabilized version of *Dm*PLK4^SBM-mCherry (*Rogers et al., 2009*). *Dm*PLK4^SBM promoted a > 6 fold increase in the amount of *Dm*Sas4 bound to WT Ana2, but this increased binding was not observed with Ana2 S38A (*Figure 2F*). These data suggest that the increased binding of CPAP/*Dm*Sas4 to phosphorylated STIL/Ana2 is conserved between human and flies.

## PLK4 phosphorylates STIL S428 to promote centriole duplication

To test the requirement of STIL S428 phosphorylation by PLK4 for centriole biogenesis, we integrated doxycycline-inducible, siRNA-resistant, WT or S428A Myc-GFP-STIL transgenes a pre-defined genomic locus in a DLD-1 host cell line. As a control, we also generated DLD-1 cells expressing a Myc-GFP-STIL S1116A transgene, which contains a mutation at a conserved PLK4 phosphorylation site in the STIL STAN motif required for efficient binding to SAS6 (*Ohta et al., 2014*; *Moyer et al., 2015*; *Dzhindzhev et al., 2014*). WT, S1116A and S428A Myc-GFP-STIL transgenes were all expressed to similar levels in cells (*Figure 3—figure supplement 1A,B*), and at a level ~2–3 fold higher than endogenous STIL. Depletion of STIL by siRNA for 48 hours resulted in 100% of mitotic cells with ≤2 centrioles, and this effect was almost completely rescued by expression of the WT STIL transgene (*Figure 3A*, *Figure 3—figure supplement 1C*) without promoting substantial centriole overduplication. By contrast, expression of either STIL S428A or S1116A only led to a partial rescue of centriole duplication (61% S428A cells and 70% S1116A cells contain ≤2 centrioles in mitosis) (*Figure 3A*). Importantly, preventing S428 phosphorylation did not affect the ability of STIL to bind to or stimulate PLK4 kinase activity, suggesting that a failure to activate PLK4 kinase activity is not responsible for the failure of centriole duplication (*Figure 3—figure supplement 2A*) (*Moyer et al., 2015*). These data show that phosphorylation of STIL S428 by PLK4 promotes centriole duplication.

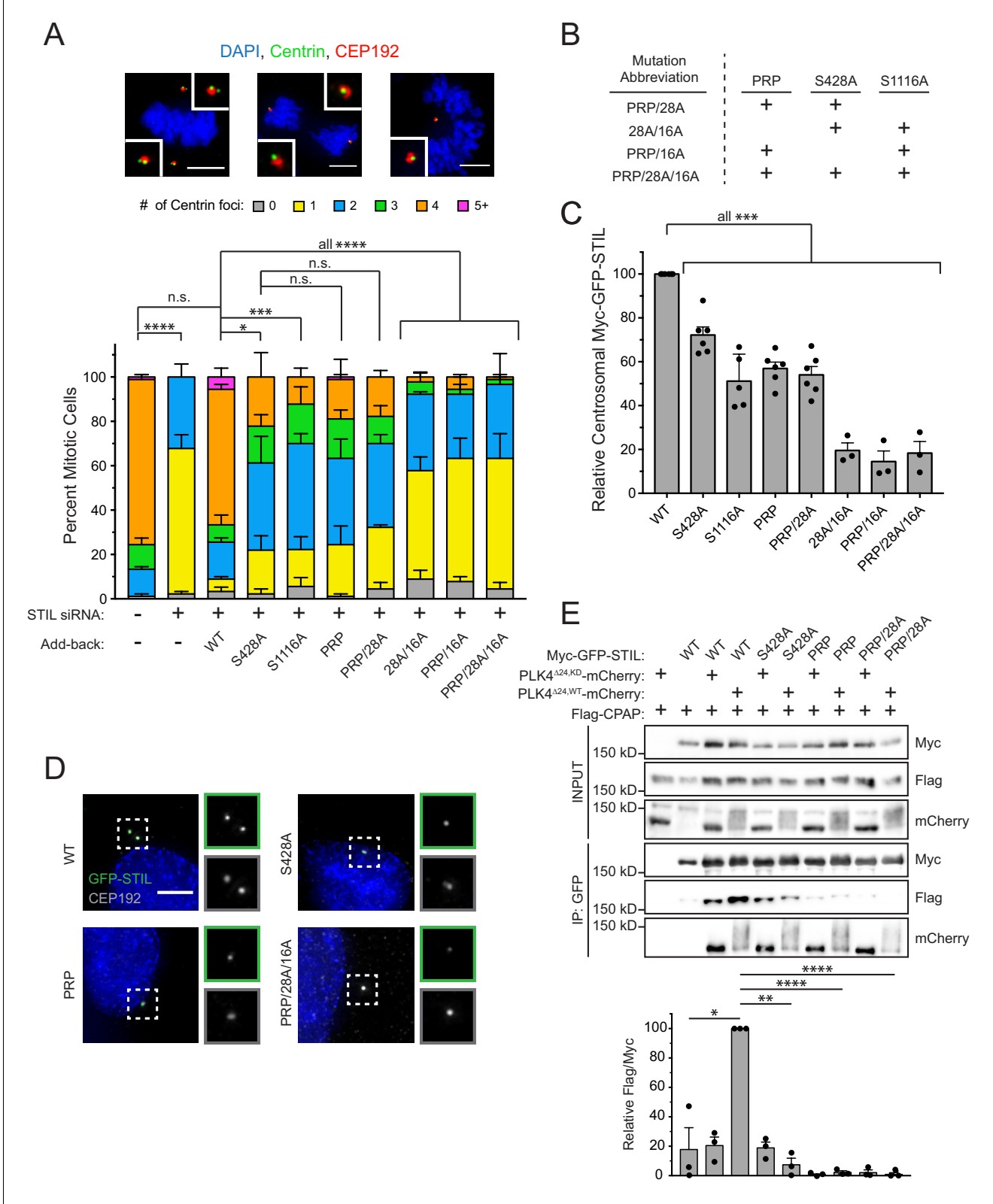

**Figure 3.** STIL S428 phosphorylation promotes centriole duplication. (**A**) Quantification showing the number of mitotic Centrin foci in cells that were depleted of endogenous STIL and induced to express a siRNA-resistant Myc-GFP-STIL transgene as indicated. Error bars represent the standard error of the mean across three independent experiments. Representative images show example mitotic cells. Scale bars represent 5 µm. (**B**) The table shows abbreviations for Myc-GFP-STIL transgenes with multiple mutations. 'PRP' represents a triple mutation of P404A, R405A, and P408A. (**C**) Quantification

*Figure 3 continued on next page*

*Figure 3 continued*

of the relative centrosomal levels of Myc-GFP-STIL constructs from (**A**) in S/G2 phase cells with at least 40 cells measured per experiment. Bars represent the mean of at least three independent experiments with the average within each experiment shown as a dot. Error bars represent the standard error of the mean. (**D**) Representative images of data shown in (**C**). Scale bar represents 5 µm. (**E**) HEK293FT cells were transfected with the indicated constructs and subjected to co-immunoprecipitation and immunoblotted with the indicated antibodies. Graph represents the mean of relative levels of immunoprecipitated Flag/Myc signal across three independent experiments. A dot displays measurement from each experiment. Error bars represent the standard error of the mean. Asterisks indicate statistically significant differences between measurements (*: $p < 0.05$; **: $p < 0.005$; ***: $p < 0.0005$; ****: $p < 0.0001$). For *Figure 3A*, statistics were calculated using an unpaired t-test against the fraction of cells containing less than four centrioles in mitosis. For *Figure 3C and E*, statistics were calculated using a one-sample t-test where mean values were tested as being different from a value of 100.

DOI: https://doi.org/10.7554/eLife.46054.007

The following source data and figure supplements are available for figure 3:

**Source data 1.** *Figure 3* Data and Statistical Analysis.
DOI: https://doi.org/10.7554/eLife.46054.012
**Source data 2.** *Figure 3—figure supplement 4* Data and Statistical Analysis.
DOI: https://doi.org/10.7554/eLife.46054.013
**Figure supplement 1.** STIL transgenes are expressed to similar levels.
DOI: https://doi.org/10.7554/eLife.46054.008
**Figure supplement 2.** The STIL PRP mutations do not affect STIL S428 phosphorylation.
DOI: https://doi.org/10.7554/eLife.46054.009
**Figure supplement 3.** Preventing STIL phosphorylation reduces the accumulation of STIL at the centriole.
DOI: https://doi.org/10.7554/eLife.46054.010
**Figure supplement 4.** Phosphorylation of the STIL STAN motif by PLK4 does not require STIL S428 phosphorylation.
DOI: https://doi.org/10.7554/eLife.46054.011

## Preventing STIL S428 phosphorylation phenocopies mutations in the CPAP binding motif of STIL

CPAP interacts with STIL via a short proline-rich region containing a highly conserved PRxxPxP motif (*Figure 2A*) (*Cottee et al., 2013*). To test whether defective CPAP binding causes the failure to rescue centriole duplication with STIL S428A, we created a PRP mutant of STIL by mutating the conserved PRPIPSP CPAP binding motif to AAPIASP (P404A, R405A, P408A). This mutation did not affect PLK4's ability to phosphorylate STIL on S428 (*Figure 3—figure supplement 2B*). As expected, the STIL PRP mutant showed impaired binding to CPAP (*Figure 3E* and *Figure 3—figure supplement 2B*). Expression of Myc-GFP-STIL PRP led to only a partial rescue of centriole duplication in cells depleted of endogenous STIL by siRNA, similar to that of the STIL S428A mutation (*Figure 3A*). Combining the S428A or PRP mutation with the S1116A mutation in the STIL STAN motif that impairs binding to SAS6 prevented any rescue of centriole duplication (*Figure 3A,B*). By contrast, a Myc-GFP-STIL transgene containing both the S428A and PRP mutation rescued centriole duplication to a level similar to that observed with a STIL transgene that contained either mutation on its own (*Figure 3A,B*). Collectively, these data suggest that S428 phosphorylation and the PRxxPxP motif of STIL function in the same pathway to promote CPAP binding to STIL, and that they both are in a separate pathway from mutations that disrupt SAS6 binding to STIL.

To evaluate the impact of disrupting CPAP and SAS6 binding on the centriole targeting of STIL, we measured the levels of Myc-GFP-STIL transgenes at the centriole in S/G2 phase cells depleted of endogenous STIL. Preventing phosphorylation at S428 reduced the localization of Myc-GFP-STIL by 28%, compared to that of the WT transgene (*Figure 3C,D*). Preventing phosphorylation at S1116 in the STAN motif reduced the abundance of STIL at the centriole by ~2 fold, as previously reported (*Figure 3C*) (*Moyer et al., 2015*). However, preventing phosphorylation of both sites reduced the centriole localization of STIL by 80%, suggesting that stable incorporation of STIL into the centriole requires strong binding to both CPAP and SAS6 (*Figure 3B,C,D*, *Figure 3—figure supplement 3*). Combining the S428A and PRP mutations did not diminish the levels of centriolar STIL below that observed with either mutation alone (*Figure 3C*). This provides further evidence that the S428A and PRP mutations function in the same pathway and act to reduce the stability of CPAP binding to STIL.

## Phosphorylation of the STIL STAN motif by PLK4 does not require STIL S428 phosphorylation

Experiments in flies showed that *Dm*PLK4 phosphorylates S38 to promote Ana2 recruitment to the centriole and then phosphorylates conserved residues in the STAN motif to enable SAS6 recruitment (*Dzhindzhev et al., 2017*; *McLamarrah et al., 2018*). However, we found that phosphorylation of STIL S428 in human cells does not abolish the centriolar recruitment of STIL (*Figure 3C,D*). To determine whether phosphorylation of the STAN motif requires phosphorylation of STIL S428, or vice versa, we monitored phosphorylation of STIL S428 and S1116 using phospho-specific antibodies. Expression of kinase active PLK4$^{\Delta24}$-mCherry promoted phosphorylation of a Myc-GFP-STIL transgene at both S428 and S1116, and mutation of either site did not prevent phosphorylation of the other (*Figure 3—figure supplement 4A*). To test if PLK4-mediated phosphorylation of the STIL STAN motif at the centriole requires phosphorylation of S428, we monitored phosphorylation of S1108 in the STAN motif using a phospho-specific antibody (*Moyer et al., 2015*). Although treatment with the PLK4 inhibitor centrinone abolished STIL S1108 phosphorylation, the S428A or PRP motif mutation did not affect phosphorylation of STIL S1108 (*Figure 3—figure supplement 4B*). These data show that phosphorylation of the STIL STAN motif by PLK4 does not require prior phosphorylation of STIL S428.

## Stable centriole recruitment of STIL requires STIL S428 phosphorylation

To examine how S428 phosphorylation affects the binding dynamics of centriolar STIL, we performed Fluorescence Recovery after Photobleaching (FRAP) in cells depleted of endogenous STIL and expressing Myc-GFP-STIL transgenes. Myc-GFP-STIL WT partially recovered following bleaching, showing that STIL exists in both mobile and immobile pool at the procentriole (*Figure 4A*, Myc-GFP-STIL WT, recovery percentage (R%)=38%). Consistent with previous observations, mutation of the S1116 phosphorylation site increased the mobile fraction of centriolar STIL (*Figure 4A*, Myc-GFP-STIL S1116A, R% = 71%) (*Moyer et al., 2015*). Importantly, the S428A and PRP mutants of STIL also showed an increased recovery of centriolar STIL (*Figure 4A*, *Figure 4—figure supplement 1*); Myc-GFP-STIL S428A, R% = 57%; Myc-GFP-STIL PRP, R% = 58%), suggesting that CPAP binding allows more stable incorporation of STIL into the procentriole.

To understand how mutations in STIL affect centrosomal CPAP dynamics, we monitored centrosomal GFP-CPAP turnover using FRAP by knocking down endogenous CPAP and expressing a siRNA-resistant Myc-GFP-CPAP transgene.

As previously reported, Myc-GFP-CPAP partially recovered after photobleaching (*Figure 4—figure supplement 2A*), Myc-GFP-CPAP, R% = 51%) (*Kitagawa et al., 2011b*). Surprisingly, performing the same measurements in cells depleted of STIL led to an almost complete turnover of Myc-GFP-CPAP (*Figure 4—figure supplement 2A*, Myc-GFP-CPAP, STIL siRNA, R% = 96%). While STIL is uniquely localized to the procentriole, CPAP is present at both the parent centriole and procentriole. In addition, CPAP has been reported to localize in the PCM material (*Sonnen et al., 2012*), consistent with a proposed role in recruiting PCM proteins (*Zheng et al., 2014*; *Gopalakrishnan et al., 2011*; *Chou et al., 2016*; *Novak et al., 2016*). Since we bleach all of the pools of CPAP in the FRAP experiments, we cannot distinguish which centrosomal populations of CPAP are dynamic and which are stably bound. Nevertheless, given STIL localizes exclusively to the procentriole, one interpretation of our data is that STIL is required for the stable incorporation of CPAP at the procentriole, while the parental centriole and PCM pool of CPAP are dynamic and display a more transient association with the centrosome.

To test the role of STIL S428 phosphorylation in modulating CPAP turnover at the centrosome, we integrated into DLD1 cells a Myc-STIL-T2A-GFP-CPAP transgene in which siRNA-resistant Myc-STIL and siRNA-resistant GFP-CPAP were both expressed from the same doxycycline-inducible promoter. As expected, Myc-STIL expression significantly suppressed the increased turnover of GFP-CPAP observed in cells depleted of endogenous CPAP and STIL (*Figure 4—figure supplement 2A*, GFP-CPAP, Myc-STIL WT background, R% = 50%). However, mutation of S428 or the PRP motif on STIL increased the turnover of GFP-CPAP compared with WT STIL (*Figure 4B* and *Figure 4—figure supplement 2B*, GFP-CPAP, Myc-STIL S428A background, R% = 69%; GFP-CPAP, Myc-STIL PRP background, R% = 67%). Expression of Myc-STIL S1116A did not increase GFP-CPAP turnover, indicating that the increase in GFP-CPAP turnover in a Myc-STIL S428A background reflects a specific

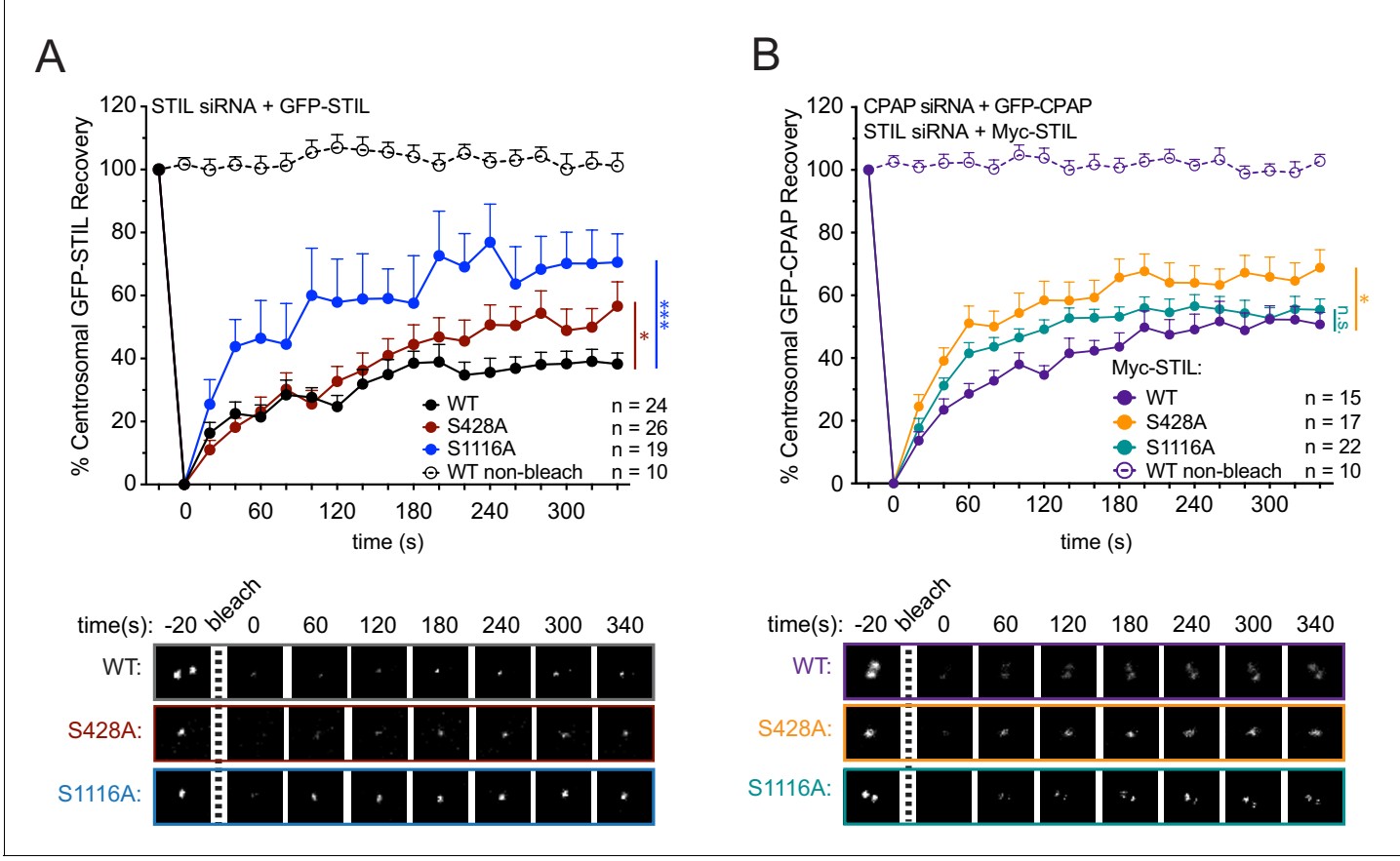

**Figure 4.** STIL S428 phosphorylation promotes the stable integration of CPAP into the centrosome. (**A**) Cells were depleted of endogenous STIL and replaced with the indicated transgene. Myc-GFP-STIL centrosomal foci were photobleached, and fluorescence recovery was measured. The number of quantified photobleaching and recovery events are indicated. Representative timepoints are shown below. Error bars represent the standard error of the mean. (**B**) Cells were depleted of endogenous STIL and CPAP and replaced with the indicated Myc-STIL-T2A-GFP-CPAP transgene. GFP-CPAP centrosomal foci were photobleached, and fluorescence recovery was measured. The number of quantified photobleaching events is indicated. Representative timepoints are shown below. Error bars represent the standard error of the mean. Asterisks indicate statistically significant differences between measurements (*: $p<0.05$; **: $p<0.005$; ***: $p<0.0005$). Statistics were calculated using an unpaired t-test against the population of recovery measurements between indicated samples at the 340 s timepoint.

DOI: https://doi.org/10.7554/eLife.46054.014

The following source data and figure supplements are available for figure 4:

**Source data 1.** *Figure 4* Data and Statistical Analysis.
DOI: https://doi.org/10.7554/eLife.46054.017
**Source data 2.** *Figure 4—figure supplement 1* Data and Statistical Analysis.
DOI: https://doi.org/10.7554/eLife.46054.018
**Source data 3.** *Figure 4—figure supplement 2* Data and Statistical Analysis.
DOI: https://doi.org/10.7554/eLife.46054.019
**Figure supplement 1.** The STIL PRP mutation mimics the turnover dynamics of the STIL S428A mutation.
DOI: https://doi.org/10.7554/eLife.46054.015
**Figure supplement 2.** Centrosomal CPAP turns over in response to STIL depletion.
DOI: https://doi.org/10.7554/eLife.46054.016

defect in the STIL/CPAP interaction (*Figure 4B*, GFP-CPAP, Myc-STIL S1116A background, R % = 55%). These data suggest that STIL binding allows a more stable incorporation of CPAP into the centrosome, possibly by facilitating interactions with CPAP at the procentriole. However, since the depletion of STIL resulted in a higher level of CPAP turnover than specifically disrupting the STIL/ CPAP interaction, it is likely STIL recruits additional proteins that collectively act to stabilize the

incorporation of CPAP into the centrosome. Together, our data show that the interaction of CPAP with STIL allows both proteins to incorporate more stably into the centrosome.

## Mutations in the CPAP TCP domain cause less stable CPAP incorporation into the centrosome

To better understand the requirement of the STIL/CPAP interaction in centriole duplication, we constructed an RNAi-replacement system in DLD-1 cells where endogenous CPAP was depleted by siRNA and replaced with near physiological levels (~2–3 fold higher) of an siRNA-resistant Myc-GFP-CPAP transgene (*Figure 5A*, *Figure 5B*). Knockdown of CPAP by siRNA resulted in 81% of mitotic cells with ≤2 centrioles, and this was largely rescued by expression of the CPAP WT transgene (*Figure 5C*) without promoting centriole overduplication. By contrast, expression of a CPAP transgene with mutations in the TCP domain that reduced binding to STIL (F1229A or E1235K), led to a partial rescue of centriole duplication (58% of F1229A cells and 48% of E1235K cells contain ≤2 centrioles in mitosis) (*Figure 5C*, *Figure 5D*) (*Cottee et al., 2013*; *Hatzopoulos et al., 2013*). Importantly, the presence of these TCP domain mutations increased the turnover of Myc-GFP-CPAP at the centrosome (*Figure 5E*, Myc-GFP-CPAP R% = 51%; Myc-GFP-CPAP F1229A, R% = 76%; Myc-GFP-CPAP E1235K, R% = 67%), but did not alter the abundance of the Myc-GFP-CPAP or STIL at the centrosome (*Figure 5F–H*). Collectively, these data support the conclusion that the STIL-CPAP interaction facilitates the stable centrosomal integration of CPAP, but that this interaction does not have a major impact on the level to which CPAP accumulates at the centrosome.

## STIL depletion reduces the localization of CPAP to the centrosome

To further test whether the overall level of CPAP present at the centrosome depends on STIL, we measured centrosomal CPAP levels in S/G2 cells depleted of STIL by siRNA. STIL knockdown reduced the level of CPAP at the centrosome by ~30% (*Figure 5—figure supplement 1A–C*). Importantly, we observed that centrosomal CPAP levels were similar following depletion of endogenous STIL and expression of a WT or mutant Myc-GFP-STIL transgene (*Figure 5—figure supplement 1D, E*). This suggests a model in which CPAP is localized to the parent centriole independently of STIL, while procentriole localized CPAP requires STIL for stable binding. However, our data argue that the role of STIL in recruiting CPAP to the centrosome is largely independent of the STIL/CPAP interaction and likely depends on the recruitment of other proteins. This is supported by our FRAP analysis which showed that depletion of STIL increased the turnover of CPAP at the centriole from 51% to 96%, while disruption of STIL binding to CPAP increased CPAP turnover to only ~70%. We conclude that the recruitment of the majority of CPAP present at the centrosome does not require CPAP binding to STIL.

## Recruitment of CPAP to de novo formed centrioles requires STIL S428 phosphorylation

The multiple populations of CPAP present at the centrosome prevented us from specifically testing the requirement of STIL S428 phosphorylation for CPAP recruitment to the procentriole. To analyze the role of STIL S428 phosphorylation in recruiting CPAP specifically to assembling procentrioles, we induced the formation of freestanding de novo centrioles in cells expressing various STIL mutants. DLD-1 cells expressing a Myc-GFP-STIL transgene were chronically treated with the PLK4 inhibitor centrinone to remove centrioles (*Wong et al., 2015*). Acentriolar cell lines were then depleted of endogenous STIL using siRNA for twenty-four hours, and centrinone was removed to induce the formation of freestanding de novo centrioles (*Figure 6A*). De novo centrioles were defined as foci marked by both PLK4 and Centrin (*Figure 6—figure supplement 1*). As expected, depletion of STIL suppressed de novo centriole assembly, and this was rescued by expression of a WT Myc-GFP-STIL transgene (48% and 87% of cells expressing WT STIL contained de novo centrioles at 24 and 72 hr after centrinone washout, respectively) (*Figure 6B*). Acentriolar cell lines expressing a S428A, S1116A, or PRP Myc-GFP-STIL transgene were all deficient in assembling de novo centrioles, with only 25%, 12%, and 33% of cells containing PLK4 and Centrin positive foci at 72 hr after centrinone removal, respectively. While cells expressing S1116A Myc-GFP-STIL formed very few GFP-STIL/PLK4 foci, the number of foci observed in cells expressing S428A and PRP Myc-GFP-STIL was comparable to that observed with the WT STIL transgene (*Figure 6C*, *Figure 6—figure supplement 1*). This

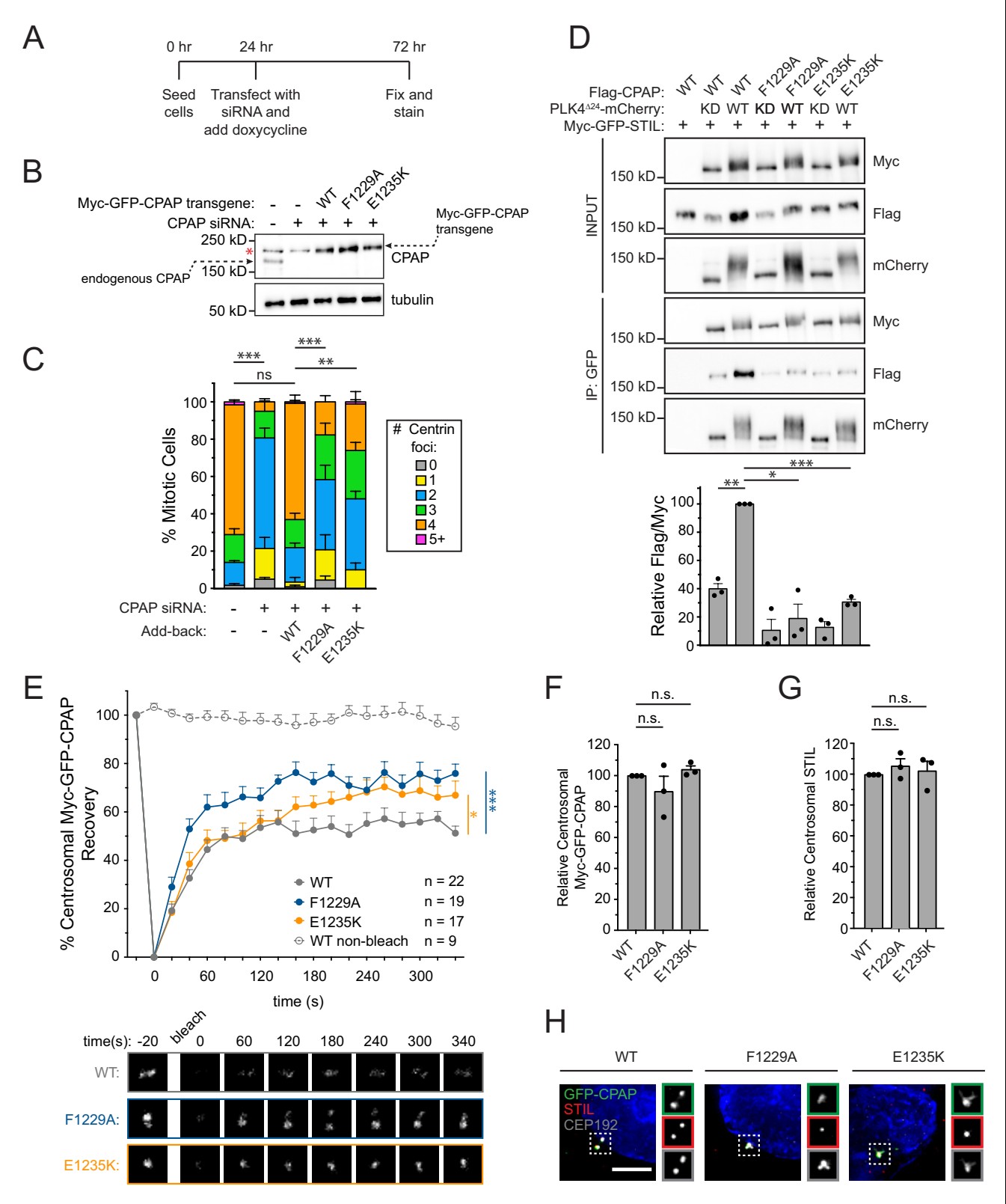

**Figure 5.** Mutations in the CPAP TCP domain cause less stable CPAP incorporation into the centrosome. (**A**) Experimental outline for the CPAP RNAi-replacement experiments. (**B**) Immunoblot showing the Myc-GFP-CPAP transgene expression levels after knockdown of endogenous CPAP. Note that a background band (denoted by a red asterisk) appears in control lanes that overlaps with the Myc-GFP-CPAP transgene. (**C**) Quantification showing the number of mitotic Centrin foci in cells depleted of endogenous CPAP and induced to express a siRNA-resistant Myc-GFP-CPAP transgene as indicated.

*Figure 5 continued on next page*

*Figure 5 continued*

Error bars represent the standard error of the mean across three independent experiments. (**D**) HEK293FT cells were transfected with the indicated constructs, subjected to co-immunoprecipitation and immunoblotted with the indicated antibodies. Graph represents the mean of relative levels of immunoprecipitated Flag/Myc signal across three independent experiments. A dot indicates the average within each experiment. Error bars represent the standard error of the mean. (**E**) Cells were depleted of endogenous CPAP and replaced with the indicated transgene. Myc-GFP-CPAP centrosomal foci were photobleached, and fluorescence recovery was measured. The number of quantified photobleaching and recovery events is indicated. Error bars represent the standard error of the mean. Representative timepoints are shown below. Note that 'WT' trace is repeated from ***Figure 4—figure supplement 2A***. (**F**) Quantification of the relative centrosomal levels of Myc-GFP-CPAP from S/G2 phase cells with at least 40 cells measured per experiment. Bars represent the mean of at least three independent experiments with the average within each experiment shown as a dot. Error bars represent the standard error of the mean. (**G**) Quantification of the relative centrosomal levels of STIL from S/G2 phase cells with at least 40 cells measured per experiment. Bars represent the mean of at least three independent experiments with the average within each experiment shown as a dot. Error bars represent the standard error of the mean. (**H**) Representative images from data quantified in (**F**) and (**G**). Scale bar represents 5 µm. Asterisks indicate statistically significant differences between measurements (*: $p<0.05$; **: $p<0.005$; ***: $p<0.0005$). For ***Figure 5C***, statistics were calculated using an unpaired t-test against the fraction of cells containing less than four centrioles in mitosis. For ***Figure 5D,F and G***, statistics were calculated using a one-sample t-test where mean values were tested as being different from a value of 100. For ***Figure 5E***, statistics were calculated using an unpaired t-test against the population of recovery measurements between indicated samples at the 340 s timepoint.

DOI: https://doi.org/10.7554/eLife.46054.020

The following source data and figure supplement are available for figure 5:

**Source data 1.** *Figure 5* Data and Statistical Analysis.
DOI: https://doi.org/10.7554/eLife.46054.022
**Source data 2.** *Figure 5—figure supplement 1* Data and Statistical Analysis.
DOI: https://doi.org/10.7554/eLife.46054.023
**Figure supplement 1.** Centrosomal CPAP localization does not depend on the STIL-CPAP interaction.
DOI: https://doi.org/10.7554/eLife.46054.021

suggests that cells expressing the S428A and PRP mutant STIL fail centriole assembly at a later stage than those expressing the S1116A mutant of STIL.

To determine why the S428A and PRP mutations fail de novo centriole formation after forming STIL/PLK4 foci, we measured the recruitment of SAS6 and CPAP to the newly formed GFP-STIL/PLK4 foci. WT Myc-GFP-STIL recruited both CPAP and SAS6 to as expected (***Figure 6D–H***). While recruitment of SAS6 was identical in WT, S428A, and PRP Myc-GFP-STIL, the S428A and PRP mutations resulted in a > 90% reduction in the amount of CPAP recruitment to the GFP-STIL/PLK4 foci (***Figure 6D–H***). Together, these data suggest that S428A and PRP mutant STIL bind to SAS6 and assemble a cartwheel but fail to recruit CPAP to complete the formation of de novo centrioles.

## Discussion

Significant progress has been made in understanding the composition of centrioles and how protein interactions can direct centriole assembly (*Andersen et al., 2003*; *Jakobsen et al., 2011*; *Firat-Karalar et al., 2014*; *Galletta et al., 2016*; *Gupta et al., 2015*). However, we have a limited understanding of which assembly steps are controlled by PLK4 to maintain the number of centrioles in cycling cells. Our data now establish that PLK4 phosphorylates its centriole substrate STIL on a conserved site close to the PRP motif to promote STIL binding to CPAP in vitro and in vivo. The STIL/CPAP complex is only the second binding interaction shown to be controlled by PLK4 and highlights a new regulated step in centriole assembly.

Our data lead us to propose a model whereby active PLK4 phosphorylates STIL at the site of procentriole assembly in two different regions with distinct functional consequences (***Figure 7***): phosphorylation of multiple residues in the STAN motif, most notably S1116, allows STIL binding to SAS6 to promote cartwheel assembly. Second, phosphorylation of S428 promotes the binding of the STIL PRP motif to CPAP, thereby linking the growing cartwheel to the triplet microtubules of the centriole wall. This model is consistent with our analysis of de novo centriole assembly, which showed that phosphorylation of the STIL STAN motif is required to recruit SAS6 to the site of procentriole assembly while phosphorylation of STIL S428 is required at a later stage to recruit CPAP to the cartwheel. Moreover, super-resolution imaging of *Drosophila* centrioles has revealed that the C-terminal region of Ana2/STIL containing the STAN motif is located closer to the cartwheel hub, while the N-terminal region containing the PRP motif is positioned close to the C-terminus of Sas-4/CPAP at the

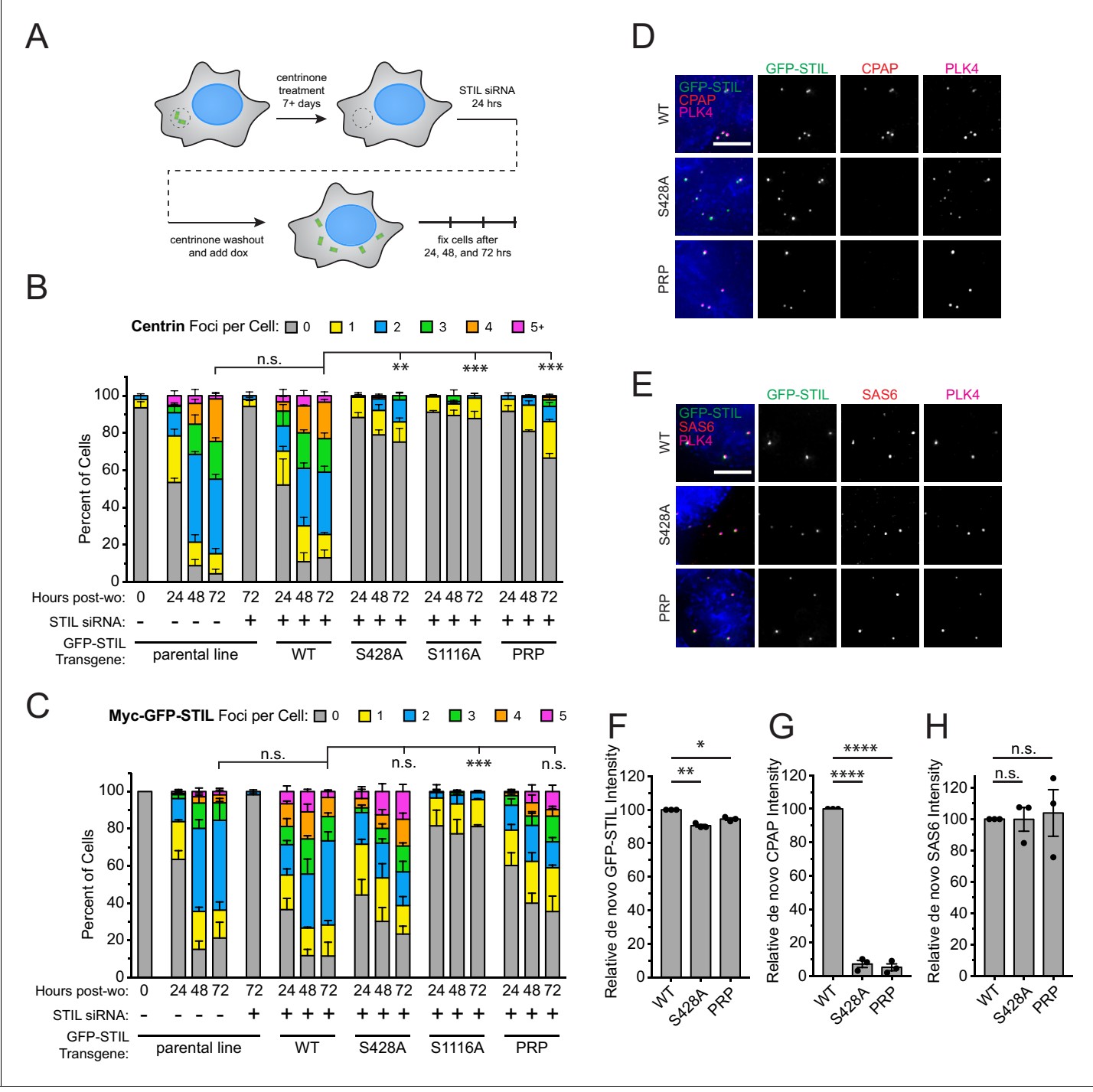

**Figure 6.** STIL S428 phosphorylation is required for recruitment of CPAP to de novo centrioles. (**A**) Experimental outline for de novo centriole assembly assay. DLD-1 cells were maintained in 500 nM centrinone for seven or more days to deplete centrioles, and then endogenous STIL was knocked down with siRNA. Twenty-four hours later, centrinone was washed out, and the expression of a siRNA-resistant Myc-GFP-STIL transgene was induced with doxycycline. Cells were analyzed at 24, 48, and 72 hr post-centrinone washout. (**B**) Quantification showing the number of Centrin foci in interphase cells under the indicated conditions. Bars represent the mean of three independent experiments in which at least 40 cells were counted per condition. Error bars represent the standard error of the mean. 'Post-wo' refers to post-washout. A Centrin focus was counted as a de novo centriole if it overlapped with a PLK4 focus (see *Figure 5—figure supplement 1*). (**C**) Quantification showing the number of Myc-GFP-STIL foci, or endogenous STIL foci in parental line, in cells under the indicated conditions. 'Post-wo' refers to post-washout. Bars represent the mean of three independent experiments in which at least 40 cells were counted per condition. Error bars represent the standard error of the mean. (**D**) Representative images of de novo centriole formation showing CPAP localization. Cells were fixed at 24 hr post-centrinone washout. Scale bar represents 5 μm. (**E**) Representative images of de

*Figure 6 continued on next page*

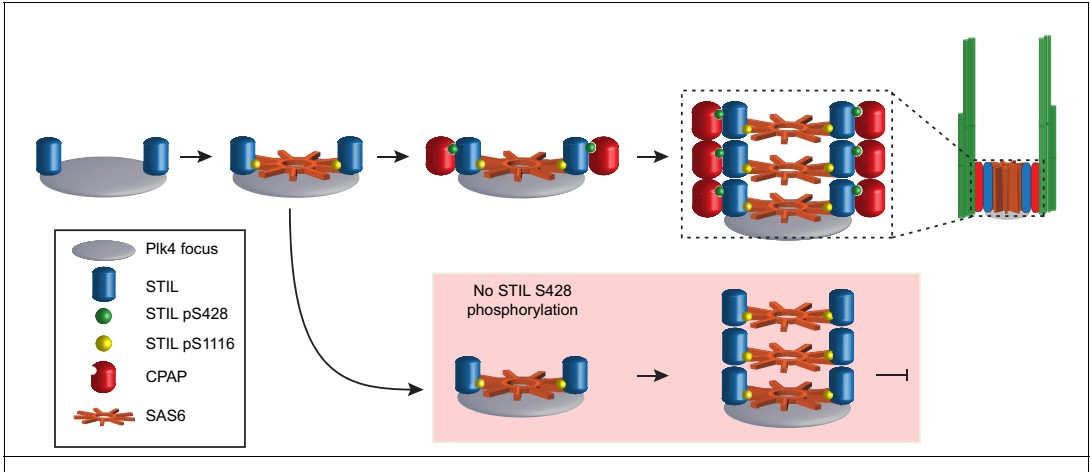

**Figure 7.** Model depicting the early stages of de novo centriole formation. A PLK4 focus forms and phosphorylates STIL at S1116 in the STAN motif to recruit SAS6 and initiate cartwheel assembly. Phosphorylation of STIL at S428 by PLK4 promotes CPAP binding and connects the growing cartwheel to the triplet microtubules of the centriole wall. In the absence of STIL S428 phosphorylation, STIL is able to recruit SAS6 and promote cartwheel assembly. However, STIL S428A fails to recruit CPAP and link the cartwheel to the microtubule wall, leading to a failure of de novo centriole assembly. In the absence of STIL S1116 phosphorylation, STIL is not effectively recruited to a PLK4 focus and cartwheel assembly fails.

DOI: https://doi.org/10.7554/eLife.46054.027

periphery of the cartwheel (*Gartenmann et al., 2017*). A mutation in the CPAP TCP domain that causes microcephaly in humans has been shown to decrease the affinity of CPAP to STIL (*Cottee et al., 2013*; *Tang et al., 2011*; *Bond et al., 2005*). Also, mutations in STIL that reside in the CPAP and SAS6 interacting motifs have also been identified in patients with microcephaly, although the significance of these alterations remains to be determined (*Cristofoli et al., 2017*).

Although recruitment of CPAP to assembling de novo centrioles requires STIL S428 phosphorylation, this modification is not required for recruiting the bulk of CPAP to the centrosome. We envisage two possible explanations for this discrepancy. First, although sharing obvious similarities, de novo centriole assembly may have some distinct requirements compared with canonical centriole biogenesis. For example, the presence of a parent centriole may help direct CPAP recruitment to the site of procentriole assembly; CPAP localized in the PCM could be recruited to the procentriole by some of CPAP's other interacting partners in the absence of STIL S428 phosphorylation. A second possibility is that multiple pools of CPAP at the centrosome (parent centriole, procentriole and PCM associated) may obscure the ability to accurately monitor the role of STIL phosphorylation in CPAP recruitment at the nascent procentriole. In any case, it is clear that even if the STIL-CPAP interaction is not strictly necessary for the recruitment of either protein to canonically duplicating centrioles, it does allow for the more stable integration of these proteins into the centrosome.

A previous study solved the structure of the CPAP TCP domain bound to a short STIL peptide (residues 395–416) containing the PRP motif but lacking the S428 phosphorylation site (*Cottee et al., 2013*). A central question that now emerges is how phosphorylation of S428, which is positioned ~20 amino acids downstream of the core PRP interaction motif in STIL, promotes binding to CPAP in cells. We envisage two, non-mutually exclusive possibilities. First, phosphorylation creates an extended binding interface that increases the affinity of STIL to CPAP. Indeed, sequence conservation in the CPAP TCP domain is not confined to the region that directly interacts with the PRP motif of STIL, but extends further along the surface of the TCP domain beta sheet, suggesting that additional contacts with STIL may occur in this region (*Cottee et al., 2013*). Moreover, there is high conservation around the S428 phosphorylation site on STIL, and this conserved motif was proposed to be well positioned to form an extended interaction interface with conserved residues in the CPAP TCP domain (*Cottee et al., 2013*). Phosphorylation of STIL S428 could, therefore, seed the binding of this conserved region and cooperatively enhance the binding of STIL to CPAP.

An alternative hypothesis is that S428 phosphorylation generates a conformational change that unmasks the PRP motif in STIL and exposes it for binding to CPAP. In support of this model, work in

*Drosophila* has shown that phosphorylation of the homologous site (S38) on Ana2/STIL leads to a dramatic mobility shift in an SDS page gel that is likely to reflect a significant conformational change in Ana2 (*Dzhindzhev et al., 2017*). It is notable that *C. elegans* SAS-5/STIL lacks an obvious PRP motif, but directly binds to SAS-4/CPAP through a disordered region (*Cottee et al., 2013*). Moreover, the SAS-4/CPAP TCP domain is required for the incorporation of SAS-4 into the centriole in *C. elegans*. SAS-5 has also been shown to bind to microtubules through a region that overlaps with the SAS-4 binding domain, suggesting that the interaction of SAS-5 with microtubules and SAS-4 may be mutually exclusive (*Bianchi et al., 2018*). In the future, it will be interesting to investigate if ZYG-1/PLK4 kinase activity regulates the critical SAS-5/SAS-4 interaction in *C. elegans* by switching SAS-5 from binding microtubules to an association with SAS-4.

# Materials and methods

## Key resources table

| Reagent type (species) or resource | Designation | Source or reference | Identifiers | Additional information |
|---|---|---|---|---|
| Cell line (*H. sapien*) | DLD-1 LacZeo | *Moyer et al., 2015*. DOI: 10.1083/jcb.201502088 | | available upon request |
| Cell line (*H. sapien*) | DLD-1 LacZeo; Myc-GFP-STIL WT | *Moyer et al., 2015*. DOI: 10.1083/jcb.201502088 | | available upon request |
| Cell line (*H. sapien*) | DLD-1 LacZeo; Myc-GFP-STIL S1116A | *Moyer et al., 2015*. DOI: 10.1083/jcb.201502088 | | available upon request |
| Cell line (*H. sapien*) | DLD-1 LacZeo; Myc-GFP-STIL S428A | this study | | DLD-1 cell line in which a Myc-GFP-STIL construct with indicated mutation is integrated at a single locus |
| Cell line (*H. sapien*) | DLD-1 LacZeo; Myc-GFP-STIL PRP | this study | | DLD-1 cell line in which a Myc-GFP-STIL construct with indicated mutation is integrated at a single locus |
| Cell line (*H. sapien*) | DLD-1 LacZeo; Myc-GFP-STIL PRP/28A | this study | | DLD-1 cell line in which a Myc-GFP-STIL construct with indicated mutation is integrated at a single locus |
| Cell line (*H. sapien*) | DLD-1 LacZeo; Myc-GFP-STIL 28A/16A | this study | | DLD-1 cell line in which a Myc-GFP-STIL construct with indicated mutation is integrated at a single locus |
| Cell line (*H. sapien*) | DLD-1 LacZeo; Myc-GFP-STIL PRP/16A | this study | | DLD-1 cell line in which a Myc-GFP-STIL construct with indicated mutation is integrated at a single locus |
| Cell line (*H. sapien*) | DLD-1 LacZeo; Myc-GFP-STIL PRP/28A/16A | this study | | DLD-1 cell line in which a Myc-GFP-STIL construct with indicated mutation is integrated at a single locus |
| Cell line (*H. sapien*) | DLD-1 LacZeo; Myc-GFP-CPAP WT | this study | | DLD-1 cell line in which a Myc-GFP-CPAP construct with indicated mutation is integrated at a single locus |

*Continued on next page*

Continued

| Reagent type (species) or resource | Designation | Source or reference | Identifiers | Additional information |
|---|---|---|---|---|
| Cell line (*H. sapien*) | DLD-1 LacZeo; Myc-GFP-CPAP F1229A | this study | | DLD-1 cell line in which a Myc-GFP-CPAP construct with indicated mutation is integrated at a single locus |
| Cell line (*H. sapien*) | DLD-1 LacZeo; Myc-GFP-CPAP E1235K | this study | | DLD-1 cell line in which a Myc-GFP-CPAP construct with indicated mutation is integrated at a single locus |
| Cell line (*H. sapien*) | DLD-1 LacZeo; Myc-STIL-T2A-GFP-CPAP | this study | | DLD-1 cell line in which a Myc-STIL-T2A-GFP-CPAP construct with indicated mutation is integrated at a single locus |
| Cell line (*H. sapien*) | DLD-1 LacZeo; Myc-STIL S428A -T2A-GFP-CPAP | this study | | DLD-1 cell line in which a Myc-STIL-T2A-GFP-CPAP construct with indicated mutation is integrated at a single locus |
| Cell line (*H. sapien*) | DLD-1 LacZeo; Myc-STIL PRP -T2A-GFP-CPAP | this study | | DLD-1 cell line in which a Myc-STIL-T2A-GFP-CPAP construct with indicated mutation is integrated at a single locus |
| Cell line (*H. sapien*) | DLD-1 LacZeo; Myc-STIL S1116A -T2A-GFP-CPAP | this study | | DLD-1 cell line in which a Myc-STIL-T2A-GFP-CPAP construct with indicated mutation is integrated at a single locus |
| Antibody | Mouse monoclonal, Myc | EMD Millipore | Cat. No. 05–724 | (WB) use 1:1000 |
| Antibody | Mouse monoclonal, Flag | Sigma | Cat. No. F1804 | (WB) use 1:1000 |
| Antibody | Rabbit polyclonal, mCherry | gift from Joo Soek-Han | | (WB) use 1:1000 |
| Antibody | Rabbit polyclonal, STIL (WB) | Bethyl | Cat. No. A302-441A | (WB) use 1:2500 |
| Antibody | Rabbit polyclonal, STIL pS428 | this study | | see Materials and methods. (WB) use 1:1000 |
| Antibody | Rabbit polyclonal, STIL pS1116 | *Moyer et al., 2015*. DOI: 10.1083/ jcb.201502088 | | (WB) use 1:1000 |
| Antibody | Rabbit polyclonal, STIL pS1108 | *Moyer et al., 2015*. DOI: 10.1083/ jcb.201502088 | | (IF) use 1:250 |
| Antibody | Mouse monoclonal, tubulin | Sigma | Cat. No. T6199 | (WB) use 1:1000 |
| Antibody | Rabbit polyclonal, Plk4 pT170 | this study | | see Materials and methods. (WB) use 1:1000 |
| Antibody | Goat polyclonal, CEP192 | this study | | see Materials and methods. (IF) use 1:1000 |
| Antibody | Rabbit polyclonal, Centrin | this study | | see Materials and methods. (IF) use 1:1000 in |

*Continued*

| Reagent type (species) or resource | Designation | Source or reference | Identifiers | Additional information |
|---|---|---|---|---|
| Antibody | Rabbit polyclonal, CPAP | gift from Karen Oegema | | (WB or IF) use 1:1000 |
| Antibody | Rabbit polyclonal, STIL (IF) | *Moyer et al., 2015*. DOI: 10.1083/jcb.201502088 | | (IF) use 1:250 |
| Antibody | Rabbit polyclonal, Plk4 | *Moyer et al., 2015*. DOI: 10.1083/jcb.201502088 | | (IF) use 1:1000 |
| Antibody | Mouse monoclonal, SAS6 | Santa Cruz | Cat. No. sc-81431 | (IF) use 1:200 |
| Chemical compound, drug | centrinone | Tocris Bioscience | Cat. No. 5687 | use at 500 nM |

## Antibody production

Human Centrin2 (a.a. 1–172) was cloned into a pET-23b bacterial expression vector (Novagen) containing an N-terminal 6xHis-SUMO1 tag. Recombinant protein was purified from *E. coli* using Ni–NTA beads (BioRad), cleaved from beads with SUMO protease and used for immunization in rabbits (ProSci). An N-terminal CEP192 fragment (a.a. 1–211) was cloned into a pGEX GST bacterial expression vector containing an N-terminal GST tag. Recombinant protein was purified from *E. coli* using glutathione sepharose beads (GoldBio), cleaved from beads with PreScission protease (GE) and used for immunization in goats. Rabbit and goat immune sera were affinity-purified using standard procedures. Affinity-purified antibodies were directly conjugated to Alexa Fluor 488, DyLight 550, or DyLight 647 fluorophores (Thermo Scientific) for use in immunofluorescence.

A synthetic phospho-peptide based on the human STIL sequence flanking serine 428 [CSVPEL(pS)LVDG] was synthesized, coupled to KLH and injected into rabbits (ProSci). Polyclonal pS428 antibodies were affinity-purified using the hSTIL phosphopeptide coupled to a SulfoLink Coupling Resin (Thermo Scientific). Additionally, a synthetic phospho-peptide based on the human PLK4 sequence flanking threonine 170 [HEKHY(pT)LCGTC] was synthesized, coupled to KLH and injected into rabbits (ProSci). Polyclonal pT170 antibodies were affinity-purified using the human PLK4 phosphopeptide coupled to a SulfoLink Coupling Resin (Thermo Scientific).

## Cell culture and drug treatments

Mammalian cell culture was performed as previously described (*Moyer et al., 2015*). Cells were maintained at 37°C in a 5% $CO_2$ atmosphere with 21% oxygen. Cells were grown in Dulbecco's Modified Eagle Medium (DMEM) containing 10% FB Essence serum (VWR), 100 U/mL penicillin, 100 U/mL streptomycin and 2 mM L-glutamine. HEK293FT cells were used in co-transfection experiments, while Flp-In TRex-DLD-1 cells (a kind gift from Stephen Taylor, the University of Manchester, UK) were used in all other experiments. Flp-In TRex-DLD-1 cells were engineered using the Flp-In TRex Core Kit (Life Technologies) to stably express the Tetracycline repressor protein and contain a single, genomic FRT/*lac*Zeo site. Centrinone (a kind gift from Karen Oegema) was dissolved in DMSO and used at a final concentration of 500 nM. DLD-1 cell lines were authenticated using STR genotyping. All cell lines were determined to be free from mycoplasma contamination using DAPI staining.

## Drosophila S2 cell culture and transfection

Drosophila S2 cells (a kind gift from Ji Hoon Kim) were cultured at room temperature in vented T-25 flasks with Schnedier media (Gibco) containing 10% Fetal Bovine Serum (Sigma). $1.5 \times 10^6$ cells were seeded in a 6-well plate in 2 mL media. The following day cells were transfected with the indicated constructs using Effectene (QIAGEN) according to the manufacturer's protocol. Cells were harvested 48 hr later and subjected to co-immunoprecipitation (procedure below).

## Cloning

Molecular cloning was performed as previously described (*Moyer et al., 2015*). All constructs were cloned into a pcDNA5/FRT/TO vector backbone (Life Technologies) and expressed from a CMV promoter. DNA constructs for Drosophila S2 transfection were cloned into a pAc vector backbone (Invitrogen).

## Generation of stable cell lines and siRNA treatment

Stable cell lines and siRNA treatment was performed as previously described (*Moyer et al., 2015*). Stable, isogenic cell lines expressing indicated constructs from a CMV promoter under the control of two Tetracycline operator sites were generated according to the manufacturer's recommendation using the FRT/Flp-mediated recombination in Flp-In TRex-DLD-1 cells (Life Technologies Flp-In TRex Core Kit). Expression of Myc-GFP-STIL was induced with 1 µg/mL Tetracycline (Sigma). Expression of Myc-GFP-CPAP was induced with 2 ng/mL Tetracycline. Expression of Myc-STIL-T2A-GFP-CPAP constructs was induced with 1 µg/mL Tetracycline. For RNA interference, $2 \times 10^5$ cells were seeded in a 6-well plate, and duplexed siRNAs were introduced using RNAiMAX (Life Technologies) 24 hr later. siRNA directed against STIL (5'-GCUCCAAACAGUUUCUGCUGGAAU-3') or CPAP (5'-AGAA UUAGCUCGAAUAGAAUU-3') was purchased from Dharmacon, and control siRNA (Universal Negative Control #1) was purchased from Sigma. 24 hr after transfection, Tetracycline was added to induce expression of RNAi-resistant Myc-GFP-STIL or Myc-STIL-T2A-GFP-CPAP. Expression of the Myc-GFP-CPAP transgene was induced concurrently with siRNA treatment. Cells were harvested and processed for immunoblotting or fixed for immunofluorescence 24 hr later.

## De novo centriole formation assay

Acentriolar Flp-In TRex-DLD-1 cell lines were generated by culturing lines in centrinone at 500 nM for seven or more days. Cells were subjected to the siRNA protocol (above). 24 hr later, centrinone was washed out where noted by replacing media twice on cells for ten minutes each and then resuspending cells across multiple coverslips and adding Tetracycline at given concentrations depending on the cell line (concentrations above). Cells were fixed at 24, 48, and 72 hr post-washout as described below.

## Co-immunoprecipitation

Co-immunoprecipitation was performed as previously described (*Moyer et al., 2015*). $2 \times 10^6$ 293 FT cells were seeded into 10 cm$^2$ dishes and 24 hr later were transfected with 3 µg of plasmid DNA. 48 hr later, transfected cells were lysed in lysis buffer [10 mM Tris pH 7.5, 0.1% Triton X-100, 250 mM NaCl, 1 mM EDTA, 50 mM NaF, 50 mM β-glycerophosphate, 1 mM DTT, 500 nM microcystin, 1 mM PMSF and EDTA-free protease inhibitor tablet (Roche)], sonicated and soluble extracts prepared. The supernatant was incubated with beads coupled to GFP-binding protein (*Rothbauer et al., 2008*). Beads were washed three times in lysis buffer, and immunopurified protein was analyzed by immunoblot.

## Immunoblotting and immunofluorescence

Immunoblotting and immunofluorescence were performed as previously described (*Moyer et al., 2015*). For immunoblot analysis, protein samples were separated by SDS-PAGE, transferred onto nitrocellulose membranes with a Trans-Blot Turbo Transfer System (BioRad) and probed with the following antibodies: tubulin (mouse DM1A anti-α-tubulin, Sigma, T6199, 1:5000), STIL (rabbit, Bethyl, A302-441A, 1:2500), FLAG M2 (mouse, Sigma, F1804, 1:1000), CPAP (rabbit, a kind gift from Karen Oegema, 1:1000), HA (rat, 3F10, Roche, 1:1000), Myc 4A6 (mouse, EMD Millipore, 1:1000), SAS6 (mouse, Santa Cruz, sc-81431, 1:1000), Plk4 pT170 (rabbit, this study, 1:1000) (*Nakamura et al., 2013*), Plk4 (rabbit, this study, 1:3200), mCherry (rabbit, a kind gift from Joo Soek-Han, 1:1000), STIL pS1116 ((*Moyer et al., 2015*), 1:500) and STIL pS428 (rabbit, this study, 1:1000). Blots were blocked with 3% BSA in PBST and washed with PBST. Phospho-antibody blots were blocked in TBS Starting Block (Thermo) supplemented with 0.05% Tween 20 and washed in TBST. Antibodies were diluted in respective blocking buffers.

For immunofluorescence, cells were grown on 18 mm glass coverslips and fixed in 100%–20C methanol for 10 min. Cells were blocked in 2.5% FBS, 200 mM glycine, and 0.1% Triton X-100 in PBS

for 1 hr. Antibody incubations were conducted in the blocking solution for 1 hr. DNA was detected using DAPI, and cells were mounted in Prolong Antifade (Invitrogen). Staining was performed with the following primary antibodies: Centrin (rabbit, directly-conjugated, this study, 1:1000), Plk4 (rabbit, directly-labeled, *Moyer et al., 2015*, 1:1000), STIL (rabbit, directly-conjugated, *Moyer et al., 2015*, 1:250), STIL pS1108 (rabbit, *Moyer et al., 2015*, 1:250), SAS6 (mouse, sc-81–431 Santa Cruz, 1:1000), CEP192 (goat, directly-labeled, this study, 1:000), CPAP-Cy3 (directly-labeled rabbit, a kind gift from Karen Oegema, Ludwig Institute for Cancer Research, CA,1:1000) and CENP-F (sheep, raised against CENP-F a.a. 1363–1640, a kind gift from Stephen Taylor, the University of Manchester, UK,1:1000). Secondary donkey antibodies were conjugated to Alexa Fluor 488, 555 or 650 (Life Technologies).

Immunofluorescence images were collected using a Deltavision Elite system (GE Healthcare) controlling a Scientific CMOS camera (pco.edge 5.5). Acquisition parameters were controlled by SoftWoRx suite (GE Healthcare). Images were collected at room temperature using an Olympus 60 × 1.42 NA or Olympus 100 × 1.4 NA oil objective at 0.2 µM z-sections and subsequently deconvolved in SoftWoRx suite. Images were acquired using Applied Precision immersion oil (N = 1.516). For quantification of signal intensity at the centrosome, deconvolved 2D maximum intensity projections were saved as 16-bit TIFF images. Signal intensity was determined using ImageJ by drawing a circular region of interest (ROI) around the centriole (ROI S). A larger concentric circle (ROI L) was drawn around ROI S. ROI S and L were transferred to the channel of interest, and the signal in ROI S was calculated using the following formula: IS – [(IL-IS/AL-AS) x AS]

A = Area, I = Integrated pixel intensity.

## Fluorescence recovery after photobleaching

Fluorescence recovery after photobleaching was performed as previously described (*Moyer et al., 2015*). Cells were seeded into 4-chamber, 35 mm glass bottom culture dishes (Greiner) and maintained in cell culture medium at 37°C and 5% $CO_2$ in an environmental control station. Images were collected using a Zeiss 40 × 1.4 NA PlanApochromat oil-immersion objective on a Zeiss LSM 780 confocal equipped with a solid-state 488 nm laser and a spectral GaAsP detector. Images were acquired using Carl Zeiss immersion oil (N = 1.518). Acquisition parameters, shutters, and focus were controlled by Zen black software (Zeiss). 11 × 0.5 µM z-sections were acquired for EGFP at each time point. Two consecutive pre-bleach scans were collected at 5% of the maximum ATOF value. Centrosome localized Myc-EGFP-STIL or EGFP-CPAP was bleached within a circular region encompassing the centrosome (~3 µM in diameter) at 100% laser power with 100 µsec dwell time. Post-bleach scans were performed at 20 s time intervals for a total period of 400 s. Maximum intensity projections were created using Zen black. The integrated intensity value within a circular region of interest in the cytosol of the cell was subtracted from an identically sized region of interest drawn around the bleached centrosome. Recovery values were plotted relative to the difference between the fluorescence pre- and post-bleach.

## Recombinant protein expression and purification

Recombinant protein expression and purification was performed as previously described (*Moyer et al., 2015*). GFP-binding protein (GBP), His-hPlk4 (a.a. 1–416), and GST-Flag-CPAP TCP domain (aa 1142–1338) were expressed and purified from *E. coli* [strain Rosetta (DE3)] using standard procedures. Recombinant GST-hSTIL was expressed and purified from SF9 insect cells (Invitrogen) using the Bac-to-Bac expression system (Invitrogen). Infected cell pellets were suspended in lysis buffer (10 mM $PO_4^{3-}$ pH 7.4, 137 mM NaCl, 2.7 mM KCl, 10% glycerol, 2 mM $MgCl_2$, 5 mM DTT, 100 nM Microcystin, 1 mM $Na_3VO_4$, 250 U of Benzonaze nuclease (Sigma), 1 mM PMSF and EDTA-free protease inhibitor tablet (Roche)) and lysed by sonication. After centrifugation at 15,000 rpm for 30 min, the supernatant was supplemented with 110 mM KCl and 0.1% Triton X-100 and incubated with Glutathione Sepharose beads (GE Healthcare) for 4 hr at 4°C. Beads were washed extensively in wash buffer [10 mM $PO_4^{3-}$ pH 7.4, 137 mM NaCl, 2.7 mM KCl, 10% glycerol, 5 mM DTT, 0.1% Triton X-100, 100 mM KCl, 1 mM PMSF and EDTA-free protease inhibitor tablet (Roche)] and protein eluted in elution buffer (10 mM $PO_4^{3-}$ pH 7.4, 137 mM NaCl, 2.7 mM KCl, 10% glycerol, with 40 mM reduced glutathione and 5 mM DTT). Protein was dialyzed into a final buffer of 10 mM

$PO_4^{3-}$ pH 7.4, 137 mM NaCl, 2.7 mM KCl, and 10% glycerol. When necessary, the GST tag was removed by overnight incubation with GST-PreScission protease (GE Healthcare).

### In vitro kinase assay

In vitro kinase assays were performed as previously described (*Moyer et al., 2015*). Assays were conducted for 30 min at 30°C in 20 mM Tris pH 7.5, 25 mM KCl, 1 mM DTT, and in the presence of 10 mM $MgCl_2$ and 100 μM ATP. 2 μg of substrate was incubated with 1 μg of His-hPlk4 (a.a. 1–416). Kinase reactions were stopped with sample buffer and analyzed by SDS-PAGE and western blotting.

### In vitro binding assay

Recombinant GST-hSTIL was bound to GSH resin (GoldBio) for four hours at 4°C in PBS, 10% glycerol, and 1 mM DTT. Beads were washed twice in kinase buffer (see above) supplemented with 10% glycerol and incubated with 6xHis-hPlk4 (a.a. 1–416) in kinase buffer with or without 10 μM cold ATP at 33°C for two hours. Reactions were then spun down and washed twice with cold binding buffer (50 mM Na-HEPES pH 7.5, 400 mM NaCl, 2 mM $MgCl_2$, 1 mM EGTA, 1 mM DTT, 0.15% Triton-X 100, 100 nM Microcystin (Calbiochem) and 0.5 mg/ml BSA) and spun in binding buffer at 4°C for 2 hr in the presence of recombinant Flag-CPAP TCP domain. Beads were washed three times in binding buffer without BSA, and then proteins were eluted in SDS sample buffer and immunoblotted.

### Mass spectrometry

Mass spectrometry was performed as previously described (*Moyer et al., 2015*). In-solution protein digestion was carried out using 'Filter Assisted Sample Preparation' (FASP) method (*Wiśniewski et al., 2009*). Data- dependent MS/MS analysis of peptides was carried out on the LTQ-Orbitrap Velos (www.thermoscientific.com) interfaced with Eksigent 2D nanoflow liquid chromatography system (www.eksigent.com system). Peptides were enriched on a 2 cm trap column (YMC gel ODS-A S-10μm), fractionated on a 75 μm x 15 cm column packed with 5 μm, 100 Å Magic AQ C18 material (Michrom Bioresources), and electrosprayed through a 15 μm emitter (PF3360-75-15-N-5, New Objective). Reversed-phase solvent gradient consisted of 0.1% formic acid with increasing levels of 0.1% formic acid, 90% acetonitrile over a period of 90 min. LTQ orbitrap Velos was set at 2.0 kV spray voltage, full MS survey scan range was set at 350–1800 m/z, data-dependent HCD MS/MS analysis set for top eight precursors with minimum signal of 2000. Other parameters include peptide isolation width of m/z 1.9; dynamic exclusion limit 30 s and normalized collision energy 35; precursor and the fragment ions resolutions were 30,000 and 15,000, respectively. Internal mass calibration was applied using lock mass ion m/z = 371.101230.

Mass spectrometry raw files were automatically processed through Proteome Discoverer 1.4 software. Raw MS and MS/MS data were isotopically resolved with deconvolution and deisotoping using Thermo Scientific Xtract and MS2-processor software in addition to default spectrum selector node. The data were searched in Refseq human entries using Mascot (v2.2.6, Matrix Sciences) search engine interfaced with different processing nodes of Proteome Discoverer 1.4. Mass tolerances on precursor and fragment masses were set to 15 ppm and 0.03 Da, respectively. Peptide validator node was used for identification confidence and 1% false discovery rate cutoff was used to filter the peptides. Phosphorylation site probability was analyzed using phosphoRS 3.0 node in Proteome discoverer software (*Taus et al., 2011*).

## Acknowledgements

We thank Randall Reed for help with the FRAP studies. We also thank Ji Hoon Kim for advice and assistance in culturing *Drosophila* S2 cells. We also would like to thank Nasser Rusan for sharing Ana2 and DmSAS6 cDNAs. We are extremely grateful to Jeremy Nathans, Antony Rosen and the Johns Hopkins Institute for Basic Biomedical Sciences for providing research support. This work was supported by the National Institutes of Health (R01GM114119), an American Cancer Society Scholar Grant (RSG-16-156-01-CCG), a National Institute of Health training grant (T32GM007445) and a National Science Foundation Graduate Research Fellowship.

## Additional information

### Funding

| Funder | Grant reference number | Author |
|---|---|---|
| National Institutes of Health | R01GM114119 | Andrew Jon Holland |
| American Cancer Society | RSG-16-156-01-CCG | Andrew Jon Holland |
| National Institutes of Health | T32GM007445 | Tyler Chistopher Moyer |

The funders had no role in study design, data collection and interpretation, or the decision to submit the work for publication.

### Author contributions

Tyler Chistopher Moyer, Conceptualization, Data curation, Writing—original draft, Writing—review and editing; Andrew Jon Holland, Conceptualization, Supervision, Funding acquisition, Methodology, Writing—original draft, Project administration, Writing—review and editing

### Author ORCIDs

Andrew Jon Holland (iD) https://orcid.org/0000-0003-3728-6367

### Decision letter and Author response

Decision letter https://doi.org/10.7554/eLife.46054.030
Author response https://doi.org/10.7554/eLife.46054.031

## Additional files

### Supplementary files

• Transparent reporting form
DOI: https://doi.org/10.7554/eLife.46054.028

### Data availability

We have included a source file for all the statistical tests performed. Source data for all of the figures have also been provided.

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
