## [Decision Letter]

Thank you for submitting your article "PLK4 promotes centriole duplication by phosphorylating STIL to link the procentriole cartwheel to the microtubule wall" for consideration by *eLife*. Your article has been reviewed by three peer reviewers, one of whom is a member of our Board of Reviewing Editors, and the evaluation has been overseen by Anna Akhmanova as the Senior Editor. The following individual involved in review of your submission has agreed to reveal their identity: Jordan W Raff (Reviewer #1).

The reviewers have discussed the reviews with one another and I have drafted this decision to help you prepare a revised submission.

Summary:

In this manuscript Moyer and Holland provide compelling evidence that PLK4 phosphorylates STIL to promote its interaction with CPAP and so to promote centriole assembly. Although a similar phosphorylation event has previously been characterised by the Glover and Rogers groups for the fly homologue of STIL (Ana2), the reviewers agreed that the consequences of this phosphorylation described here were of sufficient novelty and general interest to warrant publication in *eLife*. The reviewers raised a number of important concerns, however, that will need to be addressed prior to acceptance.

Essential revisions:

1) The relative level of expression of all the proteins tested here should be compared to the endogenous levels of expression. This information was provided in some experiments, but not in others.

2) The reasons for using the PLK4^∆24^ mutation, and for using a form of PLK4 lacking the Polo boxes (potentially altering substrate specificity) for the in vitro kinase assays need to be stated more clearly. It would also help to put the results in Supplementary file 1 into a fuller context by reporting how many of the identified phosphorylation sites match the PLK4 consensus, and how many serines and threonines were not phosphorylated (and how many of these are conserved and/or conform to the consensus).

3) Statistical analysis of the data was inconsistent and was often lacking. This led to some confusion as to what you describe as "significant". In particular your statement that "the phosphorylation of STIL S428 in human cells does not play a significant role in the centrioles recruitment of STIL" seems too strong in light of the data shown in Figure 3C (although the lack of statistics make it difficult to be sure). Statistics should be applied to all the relevant data (and the methods used fully explained in each figure legend).

4) You point out towards the end of the paper that STIL is present on only the procentriole, whereas CPAP is present on both the mother centriole and the procentriole. This is potentially important for the interpretation of some of the localisation and FRAP experiments and should be stated much earlier in the manuscript. This leads to some potential confusion. For example, you conclude that the recruitment of the majority of CPAP present at the centrosome does not require STIL. At a first glance, this might seem to overturn the conventional wisdom that STIL recruits CPAP to the assembling daughter centriole, when, in fact, your data is largely consistent with this possibility (see, for example, Figure 3F, where it seems that CPAP is localised at the older centriole normally, but is not recruited to the new centriole in the presence of the S428A mutation). Thus, your quantification of total centrosome fluorescence, rather than individual centriole fluorescence, may be somewhat misleading to readers not paying careful attention. It seems you believe that STIL is recruiting CPAP to the procentriole, but is not required to maintain CPAP at mother centrioles that have already formed (presumably prior to the depletion of STIL) – but your thinking on this only becomes clear in the Discussion. These points need to be presented more clearly. We think it would be helpful if you can provide simple schematics that illustrate your interpretation of what is occurring in the localisation and FRAP studies where this is an issue. This may require some thought.

5) The centrinone washout experiment is a good attempt to get at the requirement for STIL phosphorylation in a simplified system where new centrioles form free of a mother centriole. However, in the absence of EM evidence, we think you need to be more cautious about the nature of the various foci that are observed.

[Editors' note: further revisions were requested prior to acceptance, as described below.]

Thank you for resubmitting your work entitled "PLK4 promotes centriole duplication by phosphorylating STIL to link the procentriole cartwheel to the microtubule wall" for further consideration at *eLife*. We decided there was no need to go back to the original reviewers as we feel that there are only a small number of remaining issues that will need to be addressed prior to acceptance, outlined below:

1) You now show the levels of expression relative to the endogenous proteins for all your transgenically expressed proteins, but we think it important to state in the main text that you estimate that these are over expressed by ~2-3 fold and your arguments as to why this is not a problem.

2) You should state why the DmPLK4^SBM^ protein is stabilised and what SBM stands for.

---

## [Author Response]

Essential revisions:1) The relative level of expression of all the proteins tested here should be compared to the endogenous levels of expression. This information was provided in some experiments, but not in others.

We thank the reviewers for pointing out this omission. The expression of the Myc-GFP-STIL transgenes are shown in Figure 3—figure supplement 1A, B and is ~ 2-3-fold above that of endogenous STIL. In addition, in Figure 5B we show that the expression of the Myc-GFP-CPAP transgene is ~2-3-fold above that of endogenous CPAP. Although both the Myc-GFP-STIL and Myc-GFP-CPAP transgenes are overexpressed, expression of either transgene efficiently rescued centriole duplication failure after depletion of the respective endogenous protein and resulted in minimal centriole overduplication (Figure 3A and 5C).

2) The reasons for using the PLK4^∆24^ mutation, and for using a form of PLK4 lacking the Polo boxes (potentially altering substrate specificity) for the in vitro kinase assays need to be stated more clearly. It would also help to put the results in Supplementary file 1 into a fuller context by reporting how many of the identified phosphorylation sites match the PLK4 consensus, and how many serines and threonines were not phosphorylated (and how many of these are conserved and/or conform to the consensus).

PLK4 exists as an obligate homodimer that phosphorylates itself in trans to trigger its own degradation by the proteasome. Consequently, kinase active PLK4 is extremely unstable and difficult to express in cells. We therefore used a PLK4^∆24^ mutant that lacks the phosphorylation sites required for self-destruction. The PLK4^∆24^ mutant also accumulates to the same level independent of kinase activity, which avoids the complication of having different levels of expression for kinase active and inactive PLK4. We have added the following sentence to the text to clarify this point: “Active PLK4 triggers its own degradation and thus, we used a PLK4^∆24^ mutant that stabilizes the kinase by preventing PLK4-induced autodestruction.”

As the reviewers correctly pointed out, the in vitro kinase assays and pull-down experiments were carried out with a His-PLK4 kinase domain and not full-length recombinant PLK4. The reason for this is technical. First, full-length PLK4 is difficult to express and purify recombinantly. While we have succeeded in doing this in the past, the full-length GST-PLK4 protein is low-yield and less active than the kinase domain alone. Second, we have only succeeded in expressing PLK4 with the large solubility tag GST and in the in vitro pull-down experiments GST was used to pull-down on STIL. It was therefore more convenient to use a different affinity tag for PLK4 in this assay and we already had recombinant His-PLK4 kinase domain at hand. Finally, kinases in vitro are considered to be relatively promiscuous and therefore these assays should be treated with caution. Our interpretation is only that PLK4 *can* phosphorylate the residues we identify on STIL, but we are careful not to propose that PLK4 phosphorylates all of these sites in vivo. We have added the following sentence to the Results section to make it clear that our in vitro kinase assay used the PLK4 kinase domain and not the full-length protein: “Recombinant full-length GST-STIL was phosphorylated with the His-PLK4 kinase domain in vitro.” We also more clearly highlight that the sites we identified are considered in vitro phosphorylation sites. “Of the 84 in vitro phosphorylation sites we identified on STIL, S428 was of particular interest as it is highly conserved, matches the PLK4 consensus phosphorylation sequence and is positioned close to the known CPAP binding region on STIL…”

Finally, we have added information to Supplementary file 1 to clarify how many serine and threonine residues were not phosphorylated by His-PLK4 kinase domain and how many of these sites matched the PLK4 consensus.

3) Statistical analysis of the data was inconsistent and was often lacking. This led to some confusion as to what you describe as "significant". In particular your statement that "the phosphorylation of STIL S428 in human cells does not play a significant role in the centrioles recruitment of STIL" seems too strong in light of the data shown in Figure 3C (although the lack of statistics make it difficult to be sure). Statistics should be applied to all the relevant data (and the methods used fully explained in each figure legend).

We apologize for the confusion. We have now added statistical analysis for all of the major comparisons we make in the study. The statistical methods used are described in the figure legends and in the source manuscript file labelled ‘Statistics Summary’.

4) You point out towards the end of the paper that STIL is present on only the procentriole, whereas CPAP is present on both the mother centriole and the procentriole. This is potentially important for the interpretation of some of the localisation and FRAP experiments and should be stated much earlier in the manuscript. This leads to some potential confusion. For example, you conclude that the recruitment of the majority of CPAP present at the centrosome does not require STIL. At a first glance, this might seem to overturn the conventional wisdom that STIL recruits CPAP to the assembling daughter centriole, when, in fact, your data is largely consistent with this possibility (see, for example, Figure 3F, where it seems that CPAP is localised at the older centriole normally, but is not recruited to the new centriole in the presence of the S428A mutation). Thus, your quantification of total centrosome fluorescence, rather than individual centriole fluorescence, may be somewhat misleading to readers not paying careful attention. It seems you believe that STIL is recruiting CPAP to the procentriole, but is not required to maintain CPAP at mother centrioles that have already formed (presumably prior to the depletion of STIL) – but your thinking on this only becomes clear in the Discussion. These points need to be presented more clearly. We think it would be helpful if you can provide simple schematics that illustrate your interpretation of what is occurring in the localisation and FRAP studies where this is an issue. This may require some thought.

We agree that our presentation of these data was not optimal. We have now modified the text to more clearly present our interpretation of this data, while at the same time being careful to not to overinterpret our findings. We have also reordered the presentation of the results and added a more detailed discussion of these experiments immediately after the data is presented. In summary, our data suggests that the CPAP-STIL interaction is required for the stable incorporation of both proteins at the centrosome, but specifically disrupting this interaction doesn’t have a major impact on the degree to which either protein accumulates at the centrosome. Nevertheless, depletion of STIL does significantly increase the turnover of CPAP and decreases CPAP localization to the centrosome. This suggests that STIL recruits additional proteins that collectively act to stabilize the incorporation of CPAP into the centrosome.

To clarify our interpretation, we have added the following discussion of the CPAP turnover data: “While STIL is uniquely localized to the procentriole, CPAP is present at both the parent centriole and procentriole. […] Nevertheless, given STIL localizes exclusively to the procentriole, one interpretation of our data is that STIL is required for the stable incorporation of CPAP at the procentriole, while the parental centriole and PCM pool of CPAP are dynamic and display a transient association with the centrosome.”

We also state, “These data suggest that STIL binding allows a more stable incorporation of CPAP into the centrosome, possibly by facilitating interactions with CPAP at the procentriole. However, since the depletion of STIL resulted in a higher level of CPAP turnover than specifically disrupting the STIL/CPAP interaction, it is likely that STIL recruits additional proteins that collectively act to stabilize the incorporation of CPAP into the centrosome.”

Later in the Results, we state: “This suggests a model in which CPAP is localized to the parent centriole independently of STIL, while procentriole localized CPAP requires STIL for stable binding. […] We conclude that the recruitment of the majority of CPAP present at the centrosome does not require CPAP binding to STIL.”

Although the CPAP-STIL interaction is not required for the bulk of CPAP recruitment to the centrosome, our analysis of de novo formed centrioles shows that STIL binding to CPAP is required for the recruitment of CPAP to the site of de novo centriole formation. We present two possible explanations for this discrepancy in the Discussion: “Although recruitment of CPAP to assembling de novo centrioles requires STIL S428 phosphorylation, this modification is not required for recruiting the bulk of CPAP to the centrosome. […] In any case, it is clear that even if the STIL-CPAP interaction is not strictly necessary for the recruitment of either protein to canonically duplicating centrioles, it does allow for the more stable integration of these proteins into the centrosome.”

5) The centrinone washout experiment is a good attempt to get at the requirement for STIL phosphorylation in a simplified system where new centrioles form free of a mother centriole. However, in the absence of EM evidence, we think you need to be more cautious about the nature of the various foci that are observed.

We agree that this is an important consideration. We considered carrying out EM to confirm that the various STIL foci we observed were centrioles (for WT STIL), or centriole cartwheels (for STIL S428A). However, after consulting with an expert in the EM field, we came to the conclusion that we would not be able to detect free-standing cartwheel structures formed with the STIL S428A mutant using EM. Identifying centriole cartwheels using EM is exceedingly challenging and requires the presence of an existing centriole to direct the search area. Since we do not have EM to support our claims that the STIL foci we are observing are bone fide centrioles or free-standing cartwheels, we modified the text to state the recruitment of proteins to “STIL foci” as opposed to sites of “de novo centriole assembly”.

[Editors' note: further revisions were requested prior to acceptance, as described below.]

1) You now show the levels of expression relative to the endogenous proteins for all your transgenically expressed proteins, but we think it important to state in the main text that you estimate that these are over expressed by ~2-3 fold and your arguments as to why this is not a problem.

We thank you for your suggestion. We have added this information to the text.

2) You should state why the DmPLK4^SBM^ protein is stabilised and what SBM stands for.

We thank you for your suggestion. We have added this information to the corresponding figure legend.